# Frugivore-fruit size relationships between palms and mammals reveal past and future defaunation impacts

Jun Ying Lim [1,2✉], Jens-Christian Svenning [3], Bastian Göldel[3], Søren Faurby [4,5] & W. Daniel Kissling [1]

Mammalian frugivores are critical seed dispersers, but many are under threat of extinction. Futhermore, the impact of past and future defaunation on plant assemblages has yet to be quantified at the global scale. Here, we integrate palm and mammalian frugivore trait and occurrence data and reveal a global positive relationship between fruit size and frugivore body size. Global variation in fruit size is better explained by present-day frugivore assemblages than by Late Pleistocene assemblages, suggesting ecological and evolutionary reorganization after end-Pleistocene extinctions, except in the Neotropics, where some large-fruited palm species may have outlived their main seed dispersers by thousands of years. Our simulations of frugivore extinction over the next 100 years suggest that the impact of defaunation will be highest in the Old World tropics, and an up to 4% assemblage-level decrease in fruit size would be required to maintain the global body size–fruit size relationship. Overall, our results suggest that while some palm species may be able to keep pace with future defaunation through evolutionary changes in fruit size, large-fruited species may be especially vulnerable to continued defaunation.

[1] Institute for Biodiversity and Ecosystem Dynamics, University of Amsterdam, Amsterdam, Netherlands. [2] School of Biological Sciences, Nanyang Technological University, Singapore 637551, Singapore. [3] Section for Ecoinformatics and Biodiversity & Center for Biodiversity Dynamics in a Changing World (BIOCHANGE), Department of Biology, Aarhus University, Aarhus 8000, Denmark. [4] Department of Biological and Environmental Sciences, University of Gothenburg, Gothenburg 40530, Sweden. [5] Gothenberg Global Biodiversity Centre, Gothenburg 40530, Sweden. ✉email: junyinglim@gmail.com

Seed dispersal is a key feature in the life history of plants[1]. It benefits plants through the avoidance of seed or seedling mortality from density-dependent competition, herbivory, or predation (e.g., Janzen–Cornell effects)[2], and through the colonization of new environments or favorable recruitment sites[3]. Given this ecological and evolutionary context, many plants rely on a wide variety of frugivores for seed dispersal[1]. Between 75 and 95% of the woody species in tropical forests possess fleshy fruits[4], providing evidence for the tremendous role that frugivores play in promoting plant species richness across tropical ecosystems.

An important factor shaping plant–frugivore interactions is the relationship between the body size of frugivores and fruit size of plants[4]. Fleshy fruits predominantly dispersed by large animals tend to be larger[5,6], whereas plant species with smaller fruits are preferentially selected by smaller-sized frugivores[7]. However, while evolutionary changes in fruit or seed sizes in relation to frugivore body sizes have been demonstrated for single plant species[7], the extent to which plant–frugivore interactions have given rise to covariation in frugivore and fruit traits at the scale of whole assemblages, and over large biogeographic scales[1], remains little explored[8].

Understanding how interactions between plants and frugivores are shaped is especially challenging, as many ecosystems have undergone massive anthropogenic defaunation over thousands of years[9]. In the Late Pleistocene, extinction was strongly size-selective with mammals of larger body size being lost pre-ferentially[10], a trait-dependent pattern of extinction risk similar to modern-day mammals[11]. In places where they still survive, megafaunal mammals such as elephants or rhinoceroses, typically defined as mammals with body mass >44 kg or 100 lbs[12], play crucial and disproportionate roles in the dispersal and recruitment of trees, and thus promoting plant diversity across a range of habitats[13,14]. This is because large-bodied mammalian frugivores have large home ranges[15], greater gut capacity, and longer gut retention times, and thus can disperse seeds over long distances compared to smaller-bodied frugivores[16]. Consequently, the loss of megafauna at the end of the Pleistocene may have resulted in numerous cascading effects on their ecosystems[16,17].

Despite the role of extinct megafauna in shaping modern-day ecosystems, the degree to which the composition of present-day plant communities reflects the evolutionary and ecological legacies of past frugivore assemblages remains relatively unknown. On the one hand, some species with megafauna-adapted fruits may persist as apparent "ecological anachronisms"[5]: fruits that may have evolved for dispersal by now-extinct frugivores but are now relatively ill-suited for dispersal by the remaining present-day frugivores[6,17]. On the other hand, the ecosystem roles of Pleistocene megafauna may be in part replaced by smaller frugivores[18], and large fleshy-fruited plant species may respond by either evolving smaller fruits to accommodate the remnant frugivore community[7], or go extinct[19]. Furthermore, given the massive defaunation in the Anthropocene, a number of negative consequences for animal-dispersed plants can be expected[11,20–23], but the severity of ecological and evolutionary change such plant groups may face globally under future extinction scenarios has not been evaluated.

To study the relationship between frugivores and fleshy-fruited plants, we focus on palms (Arecaceae) and mammalian frugivores. Palms are a key component of tropical ecosystems[24,25] and an important food source for a huge variety of animals[1,26–28]. Palms show a large diversity of fruits, typically being one-seeded, and with almost all of them being adapted to seed dispersal by animals (except a few species such as the coconut, *Cocos nucifera*, and the nipa palm, *Nypa fruticans*, which are dispersed by water). Fruit sizes of palms vary widely in color, as well as size, from small fruits (0.5 cm) in some *Dypsis*, *Geonoma*, and *Chamaedorea*

species to large fruits (exceeding 10 cm in length) in genera such as *Borassus* and *Phytelephas*[24]. The seed dispersal of many palms depends on a wide variety of mammalian frugivores[27], with many species being either dispersed by megafauna[27] or possessing characteristics that suggest megafaunal mammal dispersal[6]. Furthermore, taxonomically comprehensive fruit trait data[29] and species occurrence information[30] is now readily available for palms worldwide. The palm family thus provides an ideal model system for understanding how fleshy-fruit plant assemblages may have responded to frugivore extinctions.

Here, we look at how the past, present, and future composition of mammalian assemblages shapes the relationship between palms and mammalian frugivores. We show that the size of the largest palm fruits of a given area is highly correlated with the body size of the largest frugivores in that area, highlighting the importance of large mammalian frugivores in the ecology and evolution of large-fruited palms. Furthermore, we find that variation in the fruit sizes of present-day palm communities largely reflects the body sizes of present-day frugivore assemblages. However, in the Neotropics, the body size of frugivore assemblages as they were in the Late Pleistocene explained present-day patterns in fruit size as well as the body size of current frugivores, suggesting a persistent legacy of extinct megafaunal frugivores on present-day palm assemblages. We also find that the impact of future defaunation will be greatest in the Old World tropics, with the magnitude of fruit size change necessary to maintain the fruit size–body size relationship up to twofold higher than changes in seed size documented in a well-studied Neotropical palm following historical defaunation. Given that large-fruited palms often rely on large-bodied frugivores for effective seed dispersal, and small-fruited palms are typically accessible to a greater range of seed dispersers, we conclude that large-fruited palm species may not be able to respond to defaunation through evolutionary changes in fruit size alone, and are thus especially vulnerable to the continued downsizing of frugivore assemblages.

## Results

**The fruit size–frugivore body size relationship.** To quantify the relationship between fruit and frugivore body size, we calculated the maximum palm fruit size and the maximum mammalian frugivore body size across all species occurring in "botanical countries"[31] (Fig. 1a, b). These botanical countries are standardized geographic sampling units for which complete palm species checklists are available[30]. In contrast to haphazardly collected presence-only records such as those from the Global Biodiversity Information Facility, the checklist data provide reliable and consistent, quality-checked presence–absence data for all >2500 palm species globally[30]. For mammalian frugivores, we intersected global geographic range maps with the botanical country polygons to derive presence–absence data at the same scale. Maximum values for both fruit and frugivore body size were calculated as the 95th percentile in each botanical country to reduce the influence of outliers. We examine the relationship between fruit and frugivore sizes using the maximum values (95th percentile) in each botanical country as we expect large fruits to be especially dependent upon the presence of large frugivores. We additionally analyzed the median values of fruit and frugivore body size, as trait covariation between fruits and frugivores may be shaped by indirect interactions across the entire palm and frugivore community. We, however, hypothesize that a relationship between median frugivore and median fruit size may be obscured because smaller palm fruits are also dispersed by non-mammal frugivores such as birds.

To examine the potential drivers of patterns of fruit size at the global scale, we used a model-averaging approach with ordinary

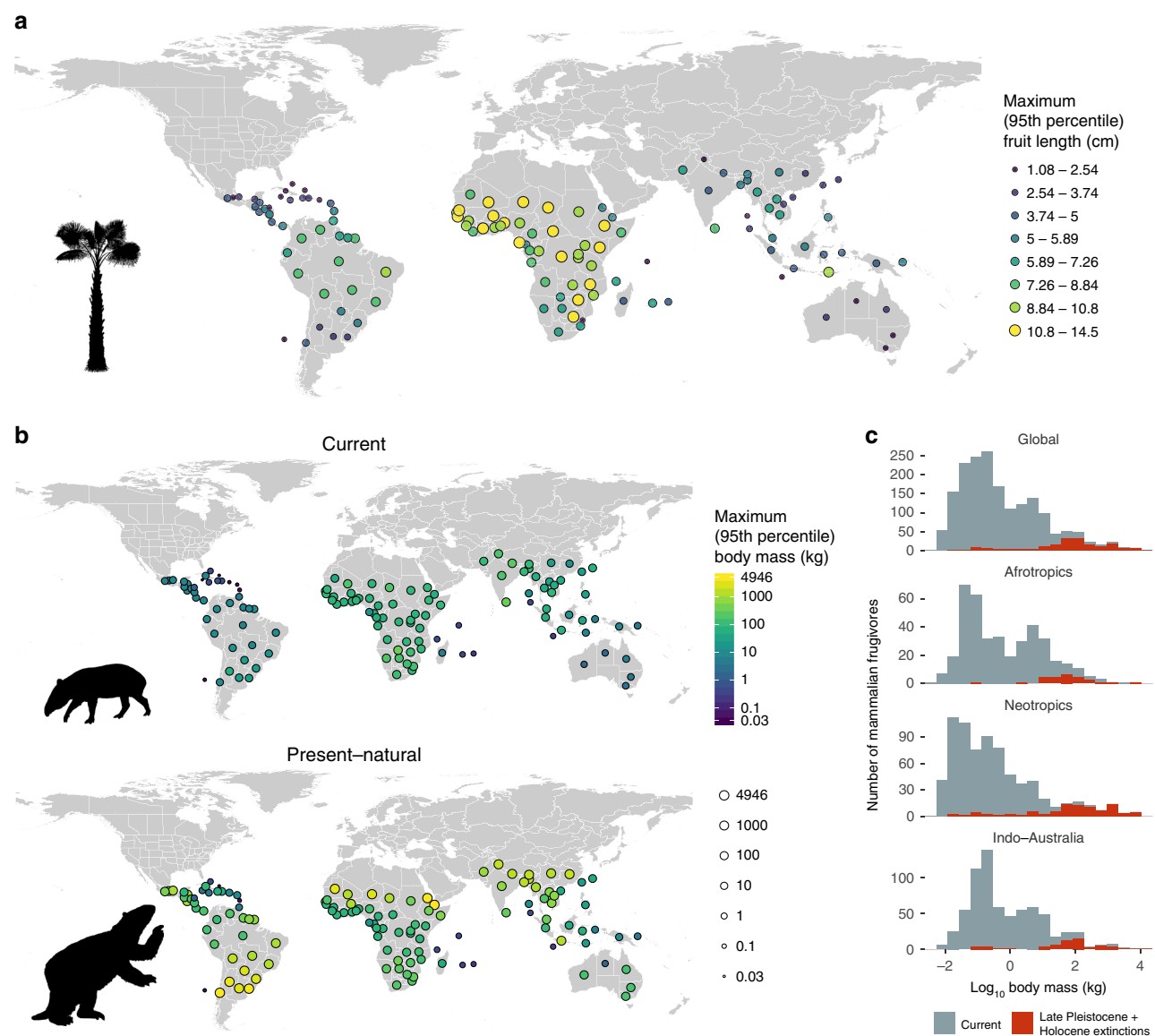

**Fig. 1 Global distribution of palm fruit sizes and mammalian frugivore body sizes.** Maps show maximum (95th percentile) values of **a** palm fruit sizes, and **b** mammal frugivore body sizes across all species within botanical countries. The "current" scenario is based on present-day distributions of extant mammalian frugivores, whereas the "present–natural" scenario is based on the distribution of all mammalian frugivores since the Last Interglacial (130 Kya), including all extant species as well as taxa that have gone extinct within that time frame. Values are represented by a colored circle plotted at the centroid of each botanical country, with brighter colors and larger circles representing higher values. In **c**, frequency distributions of body sizes of extant frugivorous mammals (gray) as well as frugivorous mammals that went extinct at the end Pleistocene and Holocene (red) are illustrated globally and within biogeographic realms. All organism silhouettes obtained from http://phylopic.org. We additionally acknowledge Julian Bayona (Zimices) for the silhouette of *Mylodon*, under a CC-Attribution-NonCommercial 3.0 Unported License (https://creativecommons.org/licenses/by-nc/3.0/). Source data are provided as a Source Data file.

least-square (OLS) models containing palm fruit size as a response variable, and log-transformed body size, present-day climate, and past climate change as potential explanatory variables (see Methods). Because different biogeographic regions have suffered varying degrees of end-Pleistocene megafaunal loss, we additionally analyzed botanical countries in biogeographic regions separately: Afrotropics, Neotropics, and Indo-Australia. To account for spatial autocorrelation, we also performed model averaging of spatial autoregressive models (see Methods). All predictor variables were first standardized, so effect sizes could be compared across variables. We also calculated an alternative set of model-averaged coefficients where standardized coefficients of candidate models were additionally standardized based on their

partial standard deviations prior to averaging to reduce the influence of collinearity on estimated model-averaged effect sizes[32] (see Methods).

Across both spatial and nonspatial models, we found that maximum fruit sizes of palm assemblages were positively correlated with maximum body sizes of extant frugivore assemblages globally (Fig. 2a, b; Supplementary Table 1). The global relationship also held when model averaging was performed on coefficients standardized by partial standard deviations (see Methods; Supplementary Table 2). This relationship was also evident when analyzing each biogeographic region separately, with the exception of the Afrotropics, where current climatic variables rather than frugivore body sizes explained most

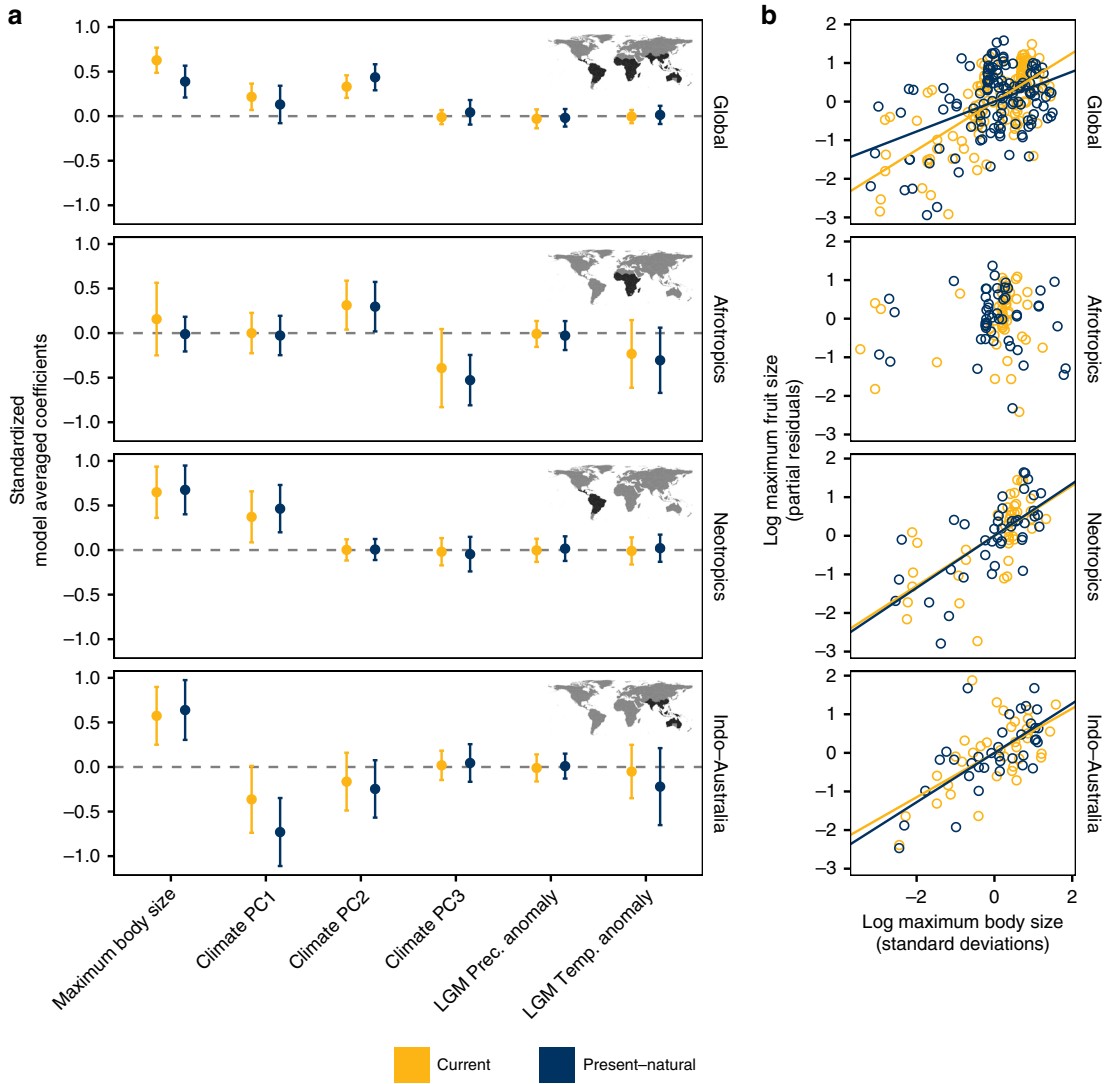

**Fig. 2 Relationship between maximum fruit sizes of palms and maximum body sizes of frugivorous mammals.** Different colors represent scenarios using current mammal assemblages (yellow) and Pleistocene assemblages that include extinct frugivores (dark blue). All maximum values are based on 95th percentiles. In **a**, relative effect sizes (standardized coefficients) of maximum body size as well as several current and paleoclimatic predictors on patterns of maximum palm fruit size at global and regional scales are indicated by dots. Bars on each dot represent 95% confidence intervals. Spatial extent of individual analyses is highlighted in inset maps through dark shading. Standardized coefficients were obtained through model averaging of ordinary least-squares (OLS) models with all possible combinations of predictor variables for each scenario. In **b**, partial residual plots show the relationships between maximum fruit and body size of mammalian frugivores globally and for each biogeographic region separately. Lines represents the model-averaged effect of maximum body size on maximum fruit size for each faunal scenario. Source data are provided as a Source Data file.

of the variation in maximum fruit sizes (Fig. 2a, Supplementary Tables 1–3).

At the global scale, climatic factors such as precipitation seasonality and mean annual temperature (represented by the second axis of a PCA; Supplementary Table 8) generally explained a smaller degree of variation in maximum fruit sizes as compared to frugivore body size, with maximum fruit size being the highest in areas with higher precipitation seasonality and (to a smaller extent) higher mean annual temperatures (Supplementary Tables 1–3). The results looking at median values were broadly similar to those using maximum values, although the relationship and explanatory power of median frugivore body size was generally weaker at both global and regional scales (Supplementary Tables 4–6).

**The impact of end-Pleistocene extinctions.** To examine whether the extinction of Pleistocene megafauna has left a persistent

signature in present-day palm assemblages, we calculated the maximum (95th percentile) and median body size of mammalian frugivores when considering all frugivores since the Last Interglacial (~130 Kya). This scenario, here defined as "present–natural", includes species that went extinct in the Late Pleistocene and Holocene[9] (see Methods), as opposed to the "current" scenario that only includes extant frugivores.

Reflecting the severity of end-Pleistocene megafaunal extinctions worldwide, average maximum (95th percentile) body sizes of frugivore assemblages globally were over 11-fold higher in the absence of end-Pleistocene and Holocene extinctions (~602 kg) than for the extant fauna (~52 kg) (Fig. 1b, c). Even with a conservative frugivore classification of extinct taxa (see Methods), average maximum body sizes were over sixfold higher (~353 kg) than today. However, while frugivore body sizes have declined globally since the end Pleistocene, the decrease in body mass has been especially dramatic in the Neotropics (Fig. 1c). While

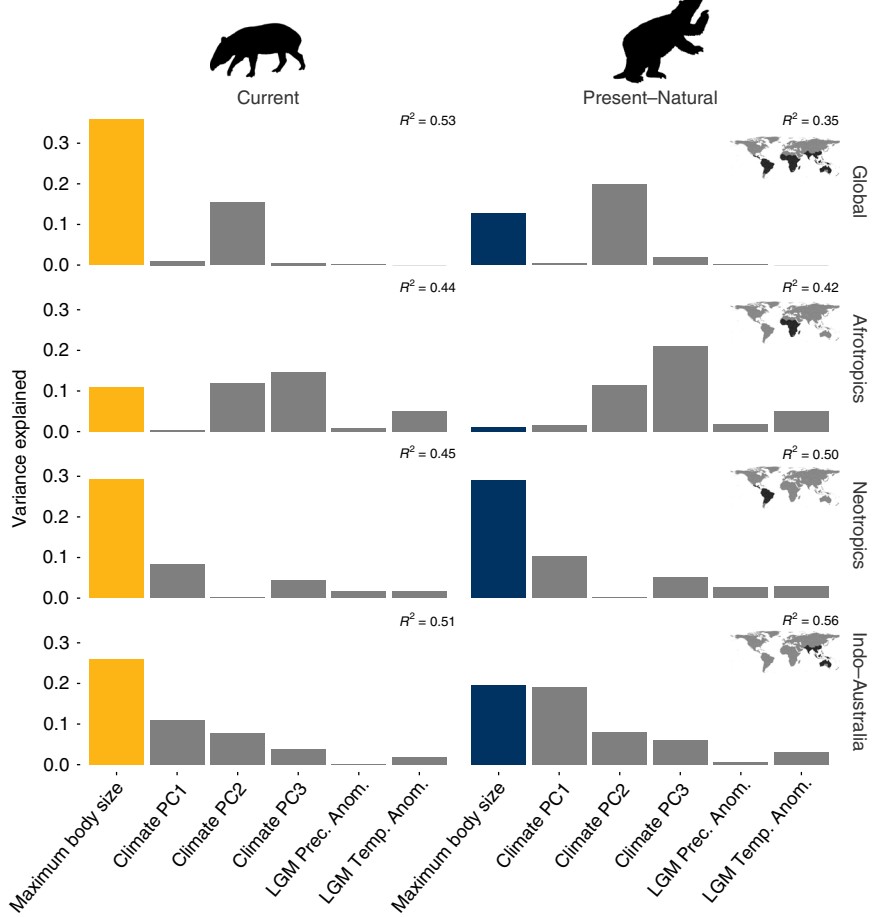

**Fig. 3 Relative importance of predictors to maximum palm fruit size under current and present–natural scenarios.** Bars represent the proportion of variance explained by each variable predictor in each model, and sum to the total variance explained by full ordinary least-squares (OLS) models with all predictor variables included ($R^2$) as indicated at the top right of each panel. The bar representing the contribution of maximum body size as a predictor is plotted for current mammal (yellow) and Pleistocene assemblages that include extinct frugivores (dark blue). Gray bars represent current and paleoclimatic variables. Spatial extent of analysis is highlighted by dark shading in inset maps. Tapir silhouette from http://phylopic.org under public domain, except for the image of *Mylodon* for which we additionally acknowledge Julian Bayona (Zimices), under a CC-Attribution-NonCommercial 3.0 Unported License (https://creativecommons.org/licenses/by-nc/3.0/). Source data are provided as a Source Data file.

Central and South America once in the past hosted up to 11 potential megafaunal mammalian frugivores with body mass over 1000 kg (such as giant sloths and various proboscideans), the largest Neotropical megafaunal frugivores today are the tapirs, *Tapirus* spp. (~150–300 kg). By contrast, megafaunal frugivores still persist in many parts of Indochina (e.g., rhinoceroses, *Rhinoceros* spp. and *Dicerorhinus sumatrensis*, and Asian elephants, *Elephas maximus*) and the Afrotropics (i.e., the African bush elephant, *Loxodonta africana*).

Globally, maximum body size of current frugivore assemblages showed a stronger relationship with present-day maximum fruit size (Fig. 2a, b) and a higher explained variance compared to present–natural frugivore assemblages (Fig. 3, Supplementary Table 1). Model-averaged OLS coefficients for models, including either current or present–natural frugivore body sizes, were 0.627 and 0.388, respectively (Fig. 2, Supplementary Table 1). Total model $R^2$ for full OLS models at the global scale containing current and present–natural frugivore body size was 53% vs. 35%, respectively (Fig. 3). The higher explanatory power of current vs. present–natural assemblages was also supported when coefficients were partially standardized before averaging (Supplementary Table 2), and under either more liberal or more conservative classifications of frugivory for extinct species (Supplementary Tables 1 and 2). Model-averaging results using spatial

autoregressive models were qualitatively similar to those of nonspatial models (Supplementary Table 3). A stronger relationship and greater explanatory power of current frugivore assemblages relative to present–natural assemblages was also generally supported when analyzing median values with both spatial and nonspatial models (Supplementary Tables 4–6).

At the regional scale, however, the body size–fruit-size relationship was generally weak in the Afrotropics even when considering present–natural assemblages. Present–natural maximum body size showed a slightly stronger relationship with current maximum fruit size within Indo-Australia than current maximum body size (Fig. 2a, b), although the proportion of variance explained by present–natural compared to current maximum body size was much lower: 26% vs. 20%, respectively. This does not appear to be driven by the severity of megafaunal extinctions in Australia and other parts of the Sahul Shelf; a reanalysis of patterns of maximum fruit size in the Indotropics, excluding botanical countries in Australia and New Guinea, yielded qualitatively similar results (Supplementary Fig. 2). Notably, present–natural Neotropical frugivore body size explained as much variation as current frugivore body size (ca. 29%, Fig. 3, Supplementary Table 1). However, present–natural body size explained substantially more variation than current body size under our spatial models: 29% and 24%, respectively

(Supplementary Table 3). The results were qualitatively similar when considering a more conservative set of putative frugivores among extinct taxa, but not under the most conservative definition (Supplementary Tables 1–3).

**Quantifying the impact of future defaunation**. To evaluate the impact of potential future mammal extinctions on palm assemblages, we calculated maximum body sizes (95th percentile) of mammal frugivore assemblages for each botanical country under future defaunation scenarios. We focused on changes in maximum body size as we expect defaunation impacts to be most pronounced among palm species with large fruits. We stochastically simulated future frugivore assemblages assuming certain probabilities that species will go extinct over the next 100 years. Extinction probabilities were derived from their IUCN Red List threat status (see Methods) using two separate approaches, which we used to generate future frugivore assemblages under defaunation scenarios of two different levels of intensity ("low" and "high"). Maximum (95th percentile) body sizes of future frugivore assemblages (mean value across 1000 simulations) were then used to generate predictions of future maximum (95th percentile) fruit size assuming the current maximum fruit size–body size relationship (see Methods). We then used the difference between predicted fruit size (assuming "defaunated" maximum body size), and present-day fruit sizes, as a measure of the defaunation impact that a particular area may face under the given defaunation scenario.

Future defaunation appears to be strongly size-selective, with extant frugivores of large body size facing higher risk of extinction (Fig. 4a). The average decline in maximum (95th percentile) frugivore body size across botanical countries in the simulations was approximately 1.4 kg under our "low" defaunation scenario, but up to 9.1 kg under our "high" defaunation scenario. This corresponds to a 0.8–15.5% decrease in the average maximum (95th percentile) body size of present-day mammal assemblages (~52 kg). To maintain the present-day global body size–fruit size relationship, maximum palm fruit sizes of future palm assemblages would have to be 0.03–0.25 cm smaller than that in present-day assemblages (Fig. 4b, c). This corresponds, on average (i.e., across species), to a 0.3–4.1% decrease in predicted maximum (95th percentile) fruit size worldwide. Furthermore, the scenarios show that global variation in the projected change in maximum (95th percentile) fruit sizes (Fig. 4b, c) would be unevenly distributed, with the largest predicted defaunation impacts concentrated in Southeast Asia, central West Africa, and some parts of the Neotropics (Fig. 4b, c).

Because 95th percentile values are dependent upon the distribution of the remaining frugivores, the 95th percentile body size of future assemblages may be higher than present-day assemblages even if decreases in true maximum body size were projected. We thus also estimated changes in true maximum fruit size under simulated changes in true maximum body size for both defaunation scenarios. Projected change in true maximum fruit size was slightly higher in magnitude (0.04–0.33 cm; Supplementary Fig. 6), although in contrast with the estimated defaunation impact using 95th percentile values, all botanical countries were now inferred to either stay the same or decrease in true maximum fruit size (Supplementary Fig. 6). Southeast Asia was similarly predicted to experience the greatest defaunation impact, although defaunation impact in the Neotropics was now projected to be the greatest in Central America, and across many parts of Northern Africa as well (Supplementary Fig. 6).

## Discussion

We found a positive relationship between palm fruit size and mammalian frugivore body size, both globally and within specific biogeographic regions. This suggests that size/trait matching in plant–frugivore interactions[8] is evident even at the scale of regional source pools (i.e., botanical countries) and despite differences in the body-size distribution of extant frugivores across biogeographic regions (Fig. 1c). Specifically, we found a strong relationship between maximum palm fruit sizes and maximum body sizes of mammalian frugivores, corroborating not only the instrumental role of large megafaunal frugivores in the evolution and ecology of large-fruited plant species[17], but also providing novel evidence that animal consumers shape broad-scale biogeographic patterns in palm fruit size. Our results do not preclude the importance of nonmammalian frugivores in the dispersal of palms, but rather highlight that large mammals disperse larger palm fruits than other frugivores such as birds[28]. They thereby contribute more strongly to patterns of maximum fruit size over large biogeographic scales. Globally, the average maximum fruit size (95th percentile) across botanical countries was approximately 6.4 cm (median 5.9 cm), which may be too large for most birds. Many birds swallow fruits as a whole and are thus constrained by gape width[7,8,33,34] and the costs associated with the physical weight and size of seeds in the gut[3,35,36]. This is especially true for palm fruits that are typically one-seeded, and which tend to have large seeds relative to their fruit sizes. Hence, large palm fruits often cannot be easily processed into smaller pieces. Large mammalian megafaunal frugivores, on the other hand, can manipulate or swallow even large fruits as a whole[27], and together with their larger gut capacity, longer gut retention times, and larger home ranges, are able to disperse the seeds of large-fruited, single-seeded plants over greater distances[16]. Some large birds (e.g., parrots and hornbills) are able to disperse some large palm fruits, but in general, such physical limitations place greater constraints on the range of fruit sizes that birds can exploit compared to mammals.

Large fruits and seeds often depend upon a small set of large-bodied frugivores[17], whereas small fruits may be eaten by frugivores from a wider range of body sizes. For instance, African elephants (*Loxodonta africana*) have been observed to disperse palm fruits ranging from the relatively small fruits of *Phoenix* (~2 cm), which are also dispersed by birds, to the large fruits of *Borassus* (~12 cm)[13,27]. Dispersal of large-fruited palms would thus be determined by the presence of the very largest frugivores present, whereas the dispersal of palms with small fruit sizes will also depend on the relative contribution of other nonmammalian frugivores. Differences in the relative accessibility of different-sized fruits to different frugivore-size classes may explain why the relationship between median fruit sizes and median body sizes of mammalian frugivore assemblages was less pronounced. However, we note that in regions where even the largest fruits are small (e.g., oceanic islands), maximum fruit size is probably not influenced by the body sizes of mammalian frugivores alone, and may also be explained by dispersal by avian frugivores.

Notably, a relationship between frugivore body size and fruit size was not evident within the Afrotropics, even though the region harbors some of the world's largest palm fruits (e.g., *Raphia* spp. and *Borassus* spp.), as well as some of the largest extant mammalian frugivores globally (Fig. 1c). The lack of a relationship at the regional scale also contrasts with observations of local ecological interactions, which show that larger palm fruits in the Afrotropics are more likely to be consumed by larger mammalian frugivores[28]. One possible explanation is that variation in maximum fruit size over larger spatial scales may be strongly constrained by climatic factors[37], as we found in our Afrotropical analysis (Fig. 2b), and is thus not primarily limited by the presence of large frugivores. The strong drying and cooling of the African continent after the Miocene may partly explain these results[38]. However, the mechanism(s) by which climate may

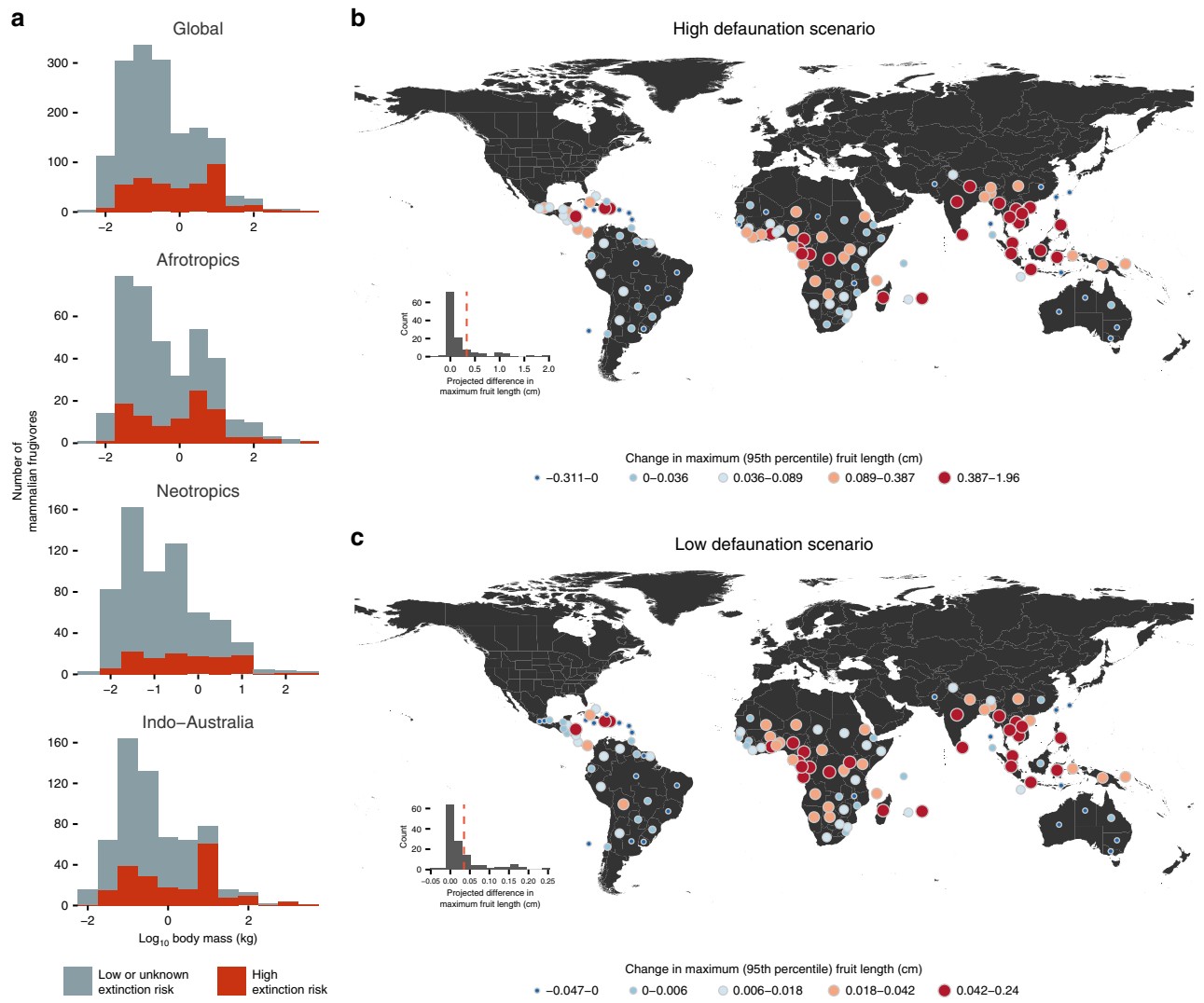

**Fig. 4 The potential impact of future defaunation on palm assemblages worldwide.** In **a**, body-size frequency distributions are shown for mammalian frugivores and colored based on IUCN Red List categories. Taxa of low or unknown extinction risk (gray) include taxa of Least Concern (LC), Near Threatened (NT) and Data Deficient (DD) IUCN Red List status, whereas taxa of high extinction risk (red) include taxa of Vulnerable (VU), Endangered (EN), or Critically Endangered (CR) IUCN red list status. Maps **b**, **c** show the impact of defaunation on palms quantified as the projected difference in maximum fruit length between present-day frugivore body sizes and frugivore body sizes under **b** high and **c** low defaunation scenarios. Values can be interpreted as the degree of ecological and evolutionary change required for palm assemblages to maintain present-day relationships between maximum frugivore body size and maximum fruit size. Values are represented by a colored circle placed at the centroid of each botanical country, with redder colors and larger circles representing greater change toward smaller fruit size. Source data are provided as a Source Data file.

control constraints on maximum fruit size remain unclear, and whether this pattern is also found in other fleshy-fruited plant families clearly deserves more empirical evidence.

Despite the strong relationship between present-day palm maximum fruit sizes and maximum frugivore body sizes, we did not find a strong imprint of Pleistocene frugivores on present-day palm fruit size. Instead, the distribution of present-day palm fruit sizes was generally better explained by present-day mammalian frugivore assemblages at both global and regional scales. Given the enormous role that Pleistocene megafaunal frugivores have likely played as seed dispersers globally[16,17,23], our results thus suggest that their extinction probably caused substantial ecological and evolutionary reorganization in palm assemblages at the global scale over thousands of years through to the present. Even when considering that some species of palm may live up to hundreds of years[24], the timescale of megafaunal extinction[39,40] still allows for potentially hundreds or thousands of generations for palm populations to respond to defaunation.

For the Neotropics, however, maximum body size of present–natural assemblages explained as much or slightly more of the variation in maximum fruit sizes than models without extinct taxa. This suggests that Neotropical palm assemblages have yet to equilibrate following end-Pleistocene frugivore extinctions, compared to other biogeographic regions (e.g., Afrotropics). This is supported by the fact that anthropogenic defaunation has been most recent and intense in the Neotropics compared to the Afrotropics and many parts of Indo-Malaya (but see Australia)[40]. Hence, Neotropical palm assemblages would not only have experienced greater perturbation through the loss of large-bodied frugivores, but also have had less time to respond to changes in frugivore assemblages since the end Pleistocene[39,40]. Simulation experiments suggest that tree species may be able to persist for up to tens of thousands of years following megafaunal frugivore extinctions[41], persisting over long periods through dispersal by smaller frugivores such as scatter-hoarding rodents and nonmammalian frugivores such as parrots[6,18,42].

Nonetheless, the loss of dispersers will result in drastic declines in recruitment success that will be difficult to overcome[42–45], perhaps even when secondary dispersers are available. Reductions in genetic diversity due to reduced gene flow from the lack of dispersers[46] may diminish the ability for plant populations to adapt to future environmental changes. We thus speculate that the long-term ecological and evolutionary responses of palm assemblages to size-selective changes in frugivore assemblages would likely involve either concomitant evolutionary changes in fruit size to cope with changes in the body-size distribution of the remaining frugivores[7], or the extinction of large-fruited plant species[19]. A recent study of seed sizes of an Amazonian palm species (*Euterpe edulis*) suggests that evolutionary changes in the order of less than 100 years are sufficient to explain present-day differences in seed size between undefaunated and defaunated forests[7]. The severe loss or extirpation of large-gape frugivorous birds may thus have forced this palm to evolve smaller fruits following defaunation. However, the small size of *E. edulis* fruits (~1.2 cm) makes it more accessible to a wider range of avian and mammalian frugivores, whereas the dispersal of the largest palm fruits may only be most effectively performed by large mammals[27,28]. As such, the dispersal of small-fruited palms is more likely to be compensated by the activities of alternative (small-bodied) frugivores compared to large-fruited palms, and small-fruited palms through diffuse and indirect coevolution with a wider range of frugivore sizes may also have the standing genetic variation necessary to evolve smaller fruit sizes.

Fruit size is not the only trait by which plants co-evolved and interact with their frugivores. Differences in fruit presentation (e.g., position of infructescence)[27], and various sensory cues such as fruit color, scent, and taste, also mediate how frugivores may perceive and use different fruits[36,47,48]. For a given species of palm, the relative dispersal effectiveness may differ among frugivores, and large frugivores may show low redundancy in fruit selection and thus may be poor ecological substitutes for each other[49,50]. For example, a study of Asian tapirs found that they were unlikely to replace elephants and rhinoceroses as they were less effective at dispersing large-seeded fruits despite their body size[51]. Thus, the degree to which plants have evolved fruit characteristics to select for different frugivores or frugivore groups will determine how likely seed dispersal will be maintained by remaining frugivores after the extinction of their primary seed dispersers. Elevated rates of extinction among large-fruited (≥4 cm) relative to small-fruited (<4 cm) palm lineages during the Quaternary in the New World but not the Old World[19], may reflect the difficulty for large-fruited palms to cope with lost mutualists. This is consistent with the greater severity of megafaunal extinctions in the New World compared to regions in the Old World like the Afrotropics[39]. Overall, the selective extinction of large-fruited palms probably played a larger role than fruit-size evolution in how palm assemblages have responded to down-sized frugivore assemblages in the aftermath of Pleistocene extinction.

The ongoing rate and magnitude of defaunation in tropical ecosystems poses enormous challenges for palms and other tropical plant groups that rely on animal-mediated seed dispersal. By simulating future defaunation, we quantified the amount of fruit-size change necessary for palm assemblages to maintain the current fruit-size frugivore body-size relationship. The magnitude of fruit-size change can be interpreted as the degree of evolutionary and ecological impact of defaunation on palm assemblages (e.g., extinction and adaptation). Using this approach, we estimate that present-day palms would require on average 0.03 cm (low defaunation scenario) to 0.25 cm (high defaunation scenario) decrease in maximum fruit size at the assemblage level to keep pace with the loss of frugivores projected over the next 100 years (Fig. 4). Furthermore, the impact of defaunation on palm assemblages

worldwide would be unevenly distributed, with the greatest future impacts expected to be in the Southeast Asian region and the Afrotropics, where many large frugivores still persist but face enormous pressure from land-use change and hunting[11,14,52]. This is consistent with the fact that the loss or decline in animal dispersers is already leading to negative effects on the demographic dynamics of Afrotropical and Southeast Asian trees[43,45,53].

Our results did not reveal a high defaunation impact on islands (except, e.g., the Mascarenes). This may reflect the small sample size of included islands ($n = 35$), the fairly conservative estimates of the probability of extinction, or the greater importance of other frugivore groups (e.g., birds and reptiles) on islands compared to continental settings[54,55]. Thus, our analysis probably underestimates the impact of future defaunation on islands, where it is additionally exacerbated by the low species richness and functional redundancy of native frugivores[46,56] and the increasing competition from invasive plant species[57]. A recent analysis of 74 tropical/subtropical islands found that extinctions of frugivores have resulted in a body mass decrease of 37% globally, and in some cases, native frugivores on islands have been completed extirpated[54]. However, it remains to be seen how effectively nonnative (alien) frugivores are compensating for the seed dispersal function provided by lost native frugivores[58], and how plant–frugivore networks will rewire given novel communities caused by climate change, translocations, and (re)introductions[59,60]. Some evidence already shows that nonnative frugivores may actually promote the dispersal of nonnative plant species more effectively than that of native plants[57], suggesting that ecosystem outcomes may be context-specific.

Nonetheless, the magnitude of estimated fruit-size change in palm assemblages is more than twofold greater than the species-level changes in palm seed size reported by Galetti et al.[7] for *E. edulis* (~0.1 cm). This is an indirect comparison (fruit vs. seed), but since most palms are one-seeded, palm fruit and seed sizes are highly correlated[7]. In addition, while there is capacity for some palm species to evolve rapidly in response to downsizing in frugivore assemblages[7], large-fruited palms likely bear a disproportionate brunt of the impact of defaunation as they are more reliant upon megafaunal frugivores compared to small-fruited palms. Hence, they may be less able to evolve to keep pace with changes in body-size composition of frugivore assemblages. Furthermore, many natural ecosystems have either already suffered the complete loss of large-vertebrate seed dispersers[21,53] or may have reduced their abundance, geographic range, or movement (e.g., through habitat fragmentation) to such a degree that the ecosystem functions they provide may be effectively lost (i.e., functional extinction)[61–64].

Although our findings paint a picture of fruit-size-dependent extinction risk at the assemblage level, it is difficult to determine the relative contribution that historic defaunation has had on documented plant extinctions[65]. Many drivers that directly shape plant extinction risk (e.g., habitat fragmentation or destruction) also contribute indirectly through their effects on seed dispersal loss. The loss of animal dispersal may result in tree-diversity declines at a local scale[45], but plant extinctions may take a long time to realize (hundreds of years) as their populations may persist long after they are functionally extinct[66]. It is thus difficult to ascertain the true magnitude of the problem. Our results suggest that the impact of defaunation will not be immediate and will likely play out over longer timescales (possibly over thousands of years), although the pace of land-use transformation and other anthropogenic impacts may accelerate this process.

## Conclusion
While some large-fruited palms may persist after the loss of their main dispersers, the loss of large frugivores will likely lead to

widespread changes in the structure and composition of ecosystems worldwide[45]. In particular, given the pace and magnitude of defaunation that modern ecosystems now face, the risk of extinction faced by palms, and perhaps other large-fruited plant groups, may be underrecognized and will play out over thousands of years even if all other anthropogenic pressures are ameliorated. We therefore urge greater consideration of the conservation of large-bodied frugivores and trophic rewilding in ecological restoration efforts[67,68]. Further research into the evolutionary and demographic consequences of different fleshy-fruited plants to defaunation[7,45], and the capacity to which interaction rewiring may occur[36,69], will be crucial for developing mitigation strategies and conservation plans for megafauna-dependent plants.

## Methods

**Palm species distribution and fruit-size data.** Species distributions for all (~2500) palm species were obtained from the World Checklist of Palms[30] (downloaded June 2015), a curated and quality-checked checklist that provides presence–absence data at the scale of "botanical countries" as defined by the International Working Group on Taxonomic Databases (TDWG, https://www.tdwg.org/)[31]. Botanical countries generally correspond to countries, but subdivide larger countries into states or provinces (e.g., United States) or retain coherent geographic units that are politically subdivided (e.g., Borneo and New Guinea) as single units[31]. We used the World Checklist of Palms rather than species occurrences (i.e., presence-only records from the Global Biodiversity Information Facility, GBIF) as the latter is taxonomically incomplete (nearly 30% of all palm species are not represented among records accessed through GBIF, downloaded January 2020). In addition, occurrence records are often geographically and taxonomically biased, resulting in large gaps and uncertainties in global occurrence information of plants (including palms)[70].

Of the 198 botanical countries in which palms occur, we included only those in the three major biogeographic realms ($n = 129$), modified from ref. [71]: (1) the Afrotropics that encompasses all of sub-Saharan Africa ($n = 51$), (2) the Neotropics ($n = 42$), which includes Central and South America and the Caribbean, and (3) Indo-Australia ($n = 36$), which includes Indo-Malaya, Australia, and Papua New Guinea. Botanical countries in other biogeographic regions (i.e., Palearctic, Nearctic, and Oceania) were excluded as they either have depauperate palm floras or are so physically isolated (e.g., Pacific islands) that dispersal filters may have disproportionately shaped both palm and frugivore assemblages. Also excluded were botanical countries for which extant mammalian frugivores are no longer present, determined by the absence of frugivorous mammals after intersecting mammal geographic range maps with the botanical country-shape file (see below). Only one botanical country was excluded for this reason—Reunion Island, where two native species of fruit bats were originally extirpated from the island during the early eighteenth century or earlier[72].

Maximum and median fruit-size values were calculated across all palm species in each botanical country. To quantify fruit size for each species, we used the PalmTraits database v1.0 (ref. [29]), which has compiled palm functional trait data of specimens from wild populations from the primary literature, monographs, species descriptions, and herbarium specimens. Specifically, we used average fruit length (rather than width or diameter) as this was the most consistently recorded fruit-size trait in the literature[29]. Maximum fruit size was defined as the 95th percentile of fruit sizes of palms in each country to reduce the influence of outliers. Because palm fruit-length information was not available for 519 palm species (~20% of global palm species richness), we filled gaps in fruit length of those species using the mean of congeneric species. This was not possible for one species, *Butyagrus nabonnandii*, a naturally occurring sterile hybrid that we thus excluded from our analysis. We additionally exclude three species from consideration—the coconut (*Cocos nucifera*) and the coco de mer (*Lodoicea maldivica*) and the nipa palm (*Nypa fruticans*)—as they have fruits that are rarely dispersed by animals[24]. Maximum and median fruit sizes across botanical countries were similar when gaps in trait data were excluded (Pearson correlation coefficient >0.99; Supplementary Fig. 1).

**Frugivore species distribution and body-size data.** To determine the effect of end-Pleistocene frugivore extinctions on present-day macroecological relationships, we defined two different faunal scenarios, representing frugivore assemblages as they are in the present day, and as they were in the Late Pleistocene—the "current" and "present–natural" scenarios following Faurby et al.[9]. The "current" scenario considers only extant frugivores, whereas the counterfactual "present–natural" scenario considers all frugivores since the Last Interglacial (130 Kya), including all extant species as well as recently extinct taxa. While climate change may have played a role in the extinction of some of these frugivores[39,73], the majority of extinctions of these species has been attributed to human arrival and activities[39,40,74,75].

"Current" and "present–natural" spatial distributions of both extant and Late Pleistocene-to-Holocene extinct mammalian frugivores were based on the

taxonomy and geographic range maps as compiled in the PHYLACINE v1.2 data set[76]. Range maps were available as Behrmann cylindrical equal area rasters, which we then overlaid with botanical country polygons to generate mammal species lists comparable to the palm presence–absence data set. Current range maps of extant mammals in the PHYLACINE data set were based on IUCN range maps (http://www.iucnredlist.org/) with minimal modification (e.g., reprojection). However, the present–natural geographic ranges of extant taxa were systematically evaluated, and modifications were made if there was strong evidence of anthropogenic impact (see Supplementary in refs. [9,76]). This also included human-mediated range expansions, which IUCN range maps for some taxa do not correct for. For extinct taxa, present–natural ranges were estimated based on fossil occurrences and the distribution of extant taxa that the extinct taxa frequently co-occurred within fossil assemblages[9,76]. This was performed by identifying grid cells that contained at least 50% of extant taxa (with a fossil record) that co-occurred with an extinct taxon in fossil assemblages/sites (data sources and further details on methodology in refs. [9,76]). Range maps with zero-occupied grid cells (e.g., species whose ranges have not been mapped by the IUCN) were excluded from our analysis. Mammal checklists for each botanical country were also manually inspected to correct errors that were sometimes introduced due to the coarse resolution of range maps (e.g., false presence of mainland taxa on islands that are closely located to mainland coastlines).

We used the MammalDIET data set[77] to identify extant species that were either primarily or secondarily frugivorous. Synonyms of binomial names in the MammalDIET data set were corrected using the IUCN Red List as a taxonomic ref. [78]. Species names that could not be reconciled with the taxonomy in the PHYLACINE data set were omitted, as were taxa whose ranges did not overlap with that of any extant palm taxa. We determined the degree to which extinct mammals were frugivorous based on indirect evidence as frugivory is not straightforwardly associated with morphological features such as dentition[79]. Recognizing this complexity, we considered three frugivore classifications with varying degrees of conservativeness. Under our "liberal" classification, we classified any extinct taxon as a frugivore if the taxon was predominantly herbivorous. This definition includes obligate grazers that may serve as seed dispersers, but probably have only a smaller fruit component in their diet compared to nongrazers. Under our "conservative" classification, we additionally included only an extinct taxon as a frugivore if it belonged to families where ≥50% of extant species are predominantly frugivorous. This definition, in contrast to the liberal classification, excludes most grazers but also many potential frugivores, largely because extant relatives may be poor analogs, since extinct relatives may vary greatly in body size and hence their feeding ecology, such as in large diprotodont (kangaroos) and pilosan (ground sloths) herbivores (Supplementary Table 7). For our intermediate "default" classification, we included taxa from families that have a borderline proportion of frugivores (≥40%). We additionally included extinct taxa that do come from families or orders that do not have extant species (e.g., litopterns and toxodons) or come from families where extant taxa may be poor analogs for diet of extinct taxa, if the available paleontological evidence (e.g., isotopic and dental microwear) suggests that the taxon may have been a browser or mixed feeder. Under this definition, ground sloths[80], litopterns[81,82], toxodons[80,83], some equids—members of the genus *Hippidion* but not *Equus*[80], some cingulids[80,84], some diprotodonts (macropodids but not Vombatidae and Diprotodontidae)[85], and murid rodents were considered putative frugivores (Supplementary Fig. 4, Supplementary Table 7).

We calculated the median and maximum frugivore body size (here defined as the 95th percentile) for each botanical country under the current scenario. For the present–natural scenario, this was performed for the three extinct frugivore classifications separately. Body-size (i.e., body mass) information was obtained from the PHYLACINE data set. In total, out of the 1930 extant frugivorous taxa in the MammalDIET data set after taxonomic reconciliation, we consider the current and present–natural ranges of 1759 and 1772 extant taxa, respectively. A total of 177 Pleistocene putative frugivores and 23 taxa extinct or extinct in the wild were additionally included in our present–natural scenario.

**Environmental data.** As fruit size may also reflect large-scale variation in climate and/or past climate change[37], the values of mean present-day climate and climate change since the Last Glacial Maximum (LGM, ~21 Kya) were calculated for each botanical country in ArcGIS (version 10.1, ESRI, Redlands, CA, USA). This was done by overlaying botanical country polygons with the corresponding climatic rasters and taking the average of overlapping raster cells. We used the CHELSA database[86] for both current and LGM climate (v1.2, 30 arc second resolution), which has advantages over the Worldclim data[87] as the CHELSA algorithm applies corrections for fine-scale orographic effects on precipitation.

For current climate, we focused on six bioclimatic variables: mean annual temperature, temperature seasonality, mean temperature of the coldest quarter, annual precipitation, precipitation seasonality, and precipitation of the driest quarter. Because some of these bioclimatic variables are highly collinear with each other, and as we were primarily interested in the relative importance of current climate and not the influence of specific climatic variables per se, we summarized the variation across the six bioclimatic variables using a principal component analysis (PCA) and used the first three principal components as climatic predictor variables. This was performed at the global scale and for each biogeographic region

("Afrotropics", "Neotropics", and "Indo-Australia") separately. The first three PCA axes ("Climate PC") explained over 90% of the variability within the bioclimatic variables among botanical countries at both global and regional scales (Supplementary Table 8).

For past climate, we calculated the magnitude of change in annual precipitation ("LGM Prec. Anom.") and mean annual temperature ("LGM Temp. Anom.") since the Last Glacial Maximum for each botanical country. This was calculated by subtracting present-day annual precipitation and mean annual temperature from the ensemble mean of mean annual precipitation and annual temperature across six different GCM models from the Paleoclimate Modeling Intercomparison Project[88] (https://pmip3.lsce.ipsl.fr/) that has been processed by the CHELSA algorithm: CCSM4, CNRM-CM5, CESS-FGOALS-g2, IPSL-CM5A-LR, MIROC-ESM, and MRI-CGCM3. Higher values thus represent a higher magnitude of climate change since the Last Glacial Maximum.

**Statistical analyses**. To evaluate the relationship between frugivore body and fruit size, we fitted both OLS linear models of both median and maximum size using present-day ("Climate PC1", "Climate PC2", and "Climate PC3") and past climate ("LGM Prec. Anom." and "LGM Temp. Anom.") as predictor variables, as well as maximum and median frugivore body size, respectively. Body-size estimates for current and present–natural scenarios, and each biogeographic region were modeled separately. Body- and fruit-size measures were log-transformed to improve normality in residuals, and all predictor variables were centered and scaled by standardizing them to unit variance.

Spatial autocorrelation was accounted for using spatial autoregressive (SAR) models. We used SAR error models (SAR$_{error}$), which have been demonstrated to be more robust than other SAR model types[89]. SAR models attempt to account for spatial autocorrelation between units within a given neighborhood structure. However, because of the irregular spacing and size of botanical countries in our study, adjacency- or distance-based neighborhoods were not appropriate. Instead, we defined the neighborhood of any given botanical country using a sphere of influence approach. The sphere of influence for each focal unit is defined as a circle of radius equal to the focal unit's distance to the centroid of its nearest neighbor. Where the sphere of the influence of two units overlaps, the two units are considered neighbors. This neighborhood definition appears to be more appropriate compared to distance-based or nearest-neighbor definitions as it takes into account the size and geometry of each botanical country. Spatial weights were row-standardized. Neighborhoods, spatial weights, and spatial SAR models were implemented using the 'spatialreg' R package v. 1.1–3 (ref. [90]), and we additionally tested for any remaining spatial autocorrelation using Moran's $I$ tests.

We used a multi-model-averaging approach to estimate effect sizes for each predictor variable across a set of candidate models[91]. The primary advantage of model averaging is that it accounts for model uncertainty across a set of candidate models, unlike more traditional stepwise model selection approaches that attempt to identify the single-best model[92,93]. We defined our candidate models as the set of ordinary least-squares (OLS) linear regression models with log-transformed palm fruit size as the response variable, with each model containing a different combination of log-transformed frugivore body size, present-day climate, and past climate change as covariates. The effect sizes for each predictor variable were then computed by averaging coefficient values across all candidate models weighted by the amount of statistical support each of these models has (i.e., the Akaike weight)[91]. To reduce the impact of model selection bias on model-averaged coefficients, the coefficient value for a predictor variable in a model where it is absent was set to zero[91,92]. This ensures that the effect sizes of predictors that are only in models with low Akaike weights are downweighted[92]. Akaike weights were calculated using AIC values that have been corrected for small sample size (AICc). Correlation between predictor variables was generally low (Supplementary Fig. 3), and variance inflation factors for all predictor variables in full OLS models (i.e., models containing all predictor variables) were mostly below 4. To reduce the influence of collinearity on model-averaging estimates, we additionally computed model-averaged coefficients where coefficients have been standardized by their partial standard deviations prior to averaging[32,93] (Supplementary Tables 2 and 5). Model averaging of both OLS and SAR models was performed using the 'dredge' and 'model.avg' functions from the 'MuMIn' R package v. 1.42.1 (ref. [94]). Standardization of coefficients of OLS models prior to model averaging was performed by setting the 'beta' argument in 'dredge' to 'partial.sd'.

In addition to model-averaged standardized effect sizes, we compare the explanatory power of body size under "current" and "present–natural" scenarios. Because the sum of Akaike weights[91] has been shown to be an inconsistent metric of variable importance[93,95], we instead calculated the proportion of variance decomposed across predictor variables[96], as implemented in the R package 'relaimpo' (v2.2–3)[97]. For simplicity, this was performed on full OLS and SAR models and not across all models tested in the model-averaging procedure. To determine the proportion of variance explained by predictor variables in the SAR models after spatial autocorrelation has been taken into account, a spatial model with just an intercept term was fitted first, and the proportion of variance in the residuals of the intercept-only spatial model explained and a full OLS model with all predictor variables was then fitted to the residuals. Given that models for current and present–natural scenarios only differ in the body-size predictor variable, we provide both $R^2$ and pseudo-$R^2$ for full OLS and SAR models under both scenarios.

**Simulating frugivore extinction**. To project fruit-size changes under a future defaunation scenario, we simulated frugivore assemblages given probabilities that species within different IUCN Red List categories will go globally extinct in the next 100 years[98,99]. We used two different sets of extinction probabilities to explore the range of magnitude of extinction risk that mammalian frugivores potentially face (Supplementary Table 9). Representing a high defaunation scenario, we used extinction probabilities derived by Davis et al.[99], who translated Red List criteria[100] for threatened categories (i.e., Critically Endangered (CR), Endangered (EN), and Vulnerable (VU)) into extinction rates for these categories, and extrapolated these estimates to derive rates for nonthreatened categories (i.e., Near Threatened (NT) and Least Concern (LC)) under the assumption that rates will increase exponentially with severity of category. Extinction probabilities ($P_{ext}$) for each category over a 100-year time frame were then calculated using these rates under a constant-rate extinction process using the following equation: $P_{ext,i} = 1 - \exp(-r_i t)$, where $r_i$ is the extinction rate for a taxon of IUCN category $i$ and $t$ equals time (Supplementary Table 9). These values are potentially high as they apply inferred estimates of extinction risk uniformly, regardless of the criteria that have been applied to a taxon, and do not explicitly consider life history variation and its effect on extinction risk. As a potential low defaunation scenario, we developed a continuous-time Markov chain (CTMC) model in the form of a transition rate matrix ($\mathbf{Q}$), the elements of which describe instantaneous rates of transition between adjacent Red List categories:

$$\mathbf{Q} = \begin{bmatrix} -- & r_{LC,NT} & 0 & 0 & 0 & 0 \\ r_{NT,LC} & -- & r_{NT,VU} & 0 & 0 & 0 \\ 0 & r_{VU,NT} & -- & r_{VU,EN} & 0 & 0 \\ 0 & 0 & r_{EN,VU} & -- & r_{EN,CR} & 0 \\ 0 & 0 & 0 & r_{CR,EN} & -- & r_{CR,EX} \\ 0 & 0 & 0 & 0 & r_{EX,CR} & -- \end{bmatrix} \quad (1)$$

where $r_{i,j}$ is the instantaneous rate of transition from IUCN category $i$ to category $j$.

We fitted our CTMC model using maximum likelihood to two data sets compiling changes (or lack thereof) in Red List categories across all IUCN-evaluated mammal species over a 12-year period[101], as well as ungulates and carnivores over a 33-year period[102]. All taxa that have been classified as Extinct in the Wild (EW), Extinct (EX), and CE species that have been flagged as possibly extinct or extinct in the wild (i.e., CR (PE) or CR (PEW) categories) were grouped into a single category representing putative extinctions. For simplicity, this category has been denoted as EX in Eq. (1). We additionally assumed that changes in Red List may only occur between adjacent ranks at any given infinitesimally small time interval, Δt. For example, for a species of LC conservation status to transition to a more severe (EN) status, it must go through (though not exclusively) statuses of intermediate severity (i.e., NT and VU). For the Di Marco et al.[102] data set, because no putatively extinct taxa were recorded to have reverted to a less severe Red List category, $r_{EX,CR}$ was set to 0 to prevent overfitting. This meant that the EX state was an absorbing state (i.e., taxa may transition into that category but not from that category). By assuming that the rates do not vary through time, the transition rate matrix can be exponentiated to give probabilities of transition between categories for any arbitrary amount of time $t$ (i.e., the transition probability matrix, $\mathbf{P}$):

$$\mathbf{P}(t) = \exp(\mathbf{Q}t) = \begin{bmatrix} P_{LC,LC}(t) & P_{LC,NT}(t) & P_{LC,VU}(t) & P_{LC,EN}(t) & P_{LC,CR}(t) & P_{LC,EX}(t) \\ P_{NT,LC}(t) & P_{NT,NT}(t) & P_{NT,VU}(t) & P_{NT,EN}(t) & P_{NT,CR}(t) & P_{NT,EX}(t) \\ P_{VU,LC}(t) & P_{VU,NT}(t) & P_{VU,VU}(t) & P_{VU,EN}(t) & P_{VU,CR}(t) & P_{VU,EX}(t) \\ P_{EN,LC}(t) & P_{EN,NT}(t) & P_{EN,VU}(t) & P_{EN,EN}(t) & P_{EN,CR}(t) & P_{EN,EX}(t) \\ P_{CR,LC}(t) & P_{CR,NT}(t) & P_{CR,VU}(t) & P_{CR,EN}(t) & P_{CR,CR}(t) & P_{CR,EX}(t) \\ P_{EX,LC}(t) & P_{EX,NT}(t) & P_{EX,VU}(t) & P_{EX,EN}(t) & P_{EX,CR}(t) & P_{EX,EX}(t) \end{bmatrix}, \quad (2)$$

where $p_{i,j}(t)$ is the probability of a taxa starting with status $i$ ending up with status $j$ after a given amount of time $t$.

We then assumed changes (or lack thereof) in Red List categories for each species to be an independent realization of a CTMC process. Thus, the likelihood of observing a given set of transition rate changes given a time frame $t$, could be calculated as the compound transition probability across species

$$\mathcal{L}(\mathbf{Q}, t | \mathbf{D}) = Pr(\mathbf{D} | \mathbf{Q}, t) = \prod^i \prod^j P_{i,j}(t) \times n_{i,j}, \quad (3)$$

where $n_{i,j}$ is the number of taxa that started in Red List category $i$ and ended in Red List category $j$ after time $t$

Rate estimates for both data sets were obtained by maximizing Eq. (3) using the 'optim' function in R. Robustness of rate estimates was tested using parametric bootstrapping (Supplementary Fig. 5). Extinction probabilities for each category (i.e., probability of transitioning into the EX category over a 100-year time period) were then calculated using the average of maximum-likelihood rate estimates for the two data sets using Eq. (2) (Supplementary Table 9). These values are potentially low estimates of extinction risk, as they are based on past changes in Red List status, whereas anthropogenic impacts may intensify into the future[14].

Extinction was stochastically simulated for all mammalian frugivores 1000 times and the maximum body sizes (95th percentile) of botanical countries was recorded. Species categorized as data deficient were conservatively assumed to have the same extinction probability as those of LC status. For simulations where all mammalian frugivores in a botanical country went extinct, maximum body size was set to zero. Mean values of maximum body sizes across simulations were then used in conjunction with global OLS models of maximum fruit size parameterized using all climatic variables and the current maximum body size of frugivore assemblages, to compute projected changes in fruit size under a defaunated scenario. This analysis was repeated using absolute maximum values of fruit and body size (Supplementary Fig. 6).

**Reporting summary**. Further information on research design is available in the Nature Research Reporting Summary linked to this article.

## Data availability

Source data are provided with this paper. Range maps and functional trait data are openly available on the Dryad Digital Repository (https://doi.org/10.5061/dryad.bp26v20). Palm checklist data are available from the World Checklist of Selected Plant Families (http://apps.kew.org/wcsp). Palm functional trait data are openly available on the Dryad Digital Repository (https://doi.org/10.5061/dryad.ts45225). Current climate and paleoclimate data are openly available online from http://www.chelsa-climate.org and on the Dryad Digital Respository (https://doi.org/10.5061/dryad.kd1d4). Data sets and derived products are also stored in a publicly available Dryad digital repository (https://doi.org/10.5061/dryad.6hdr7sqwt). Source data are provided with this paper.

## Code availability

All data sets, as well as scripts involved in data processing, statistical analysis, and plotting of the results (including a Source Data file), are published in a publicly available Dryad digital repository (https://doi.org/10.5061/dryad.6hdr7sqwt).

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

## Acknowledgements

W.D.K. acknowledges funding from the Netherlands organization for Scientific Research (824.15.007) and the University of Amsterdam (via a starting grant and through the Faculty Research Cluster "Global Ecology"). S.F. acknowledges funding from the Swedish Research Council (2017-03862). J.C.S. considers this work a contribution to his Carlsberg Foundation Semper Ardens project MegaPast2Future (grant CF16-0005), his VILLUM Investigator project "Biodiversity Dynamics in a Changing World" funded by VILLUM FONDEN (grant 16549), and his TREECHANGE project funded by Independent Research Fund Denmark—Natural Sciences (grant 6108-00078B).

## Author contributions

J.Y.L., B.G., J.-C.S., and W.D.K. conceived the idea. J.Y.L., B.G., W.D.K., and S.F. compiled the data. J.Y.L. analyzed the data. J.Y.L. and W.D.K. co-wrote the paper. All authors discussed the final results and commented on the paper.

## Competing interests

The authors declare no competing interests.
