## [Peer Review File · Nature Communications]

Reviewers' Comments:

Reviewer #1:

Remarks to the Author:

I enjoyed this a lot. It is an important question, probably the best data available to answer it, and the analysis appears to be robust and defensible. I have only a few minor comments and suggestions. First, although the focus on maximum/maximum neatly deals with the problems of palm fruit and frugivore diversity, I think readers need a brief introduction to at least the diversity of palm fruits, which vary greatly in size, external protection, color, and dispersal agents. In the defaunated sites in Asia, the largest avian frugivores - fruit pigeons and hornbills - are taking palm fruits (< 4 cm diameter) as large as the largest surviving mammals. I also think you need to be clearer on the consequences of looking at fruit sizes rather than seed size, with the former less relevant for mammals but more easy to get. I also think that the differences between regions were inadequately discussed. It is not obvious to me why regional results should differ at all.

Reviewer #2:

Remarks to the Author:

Lim et al. (NCOMMS-19-32434-T) is an interesting and well-written study that will draw more attention to the current extinction crisis and to Quaternary extinctions, which are still not very widely known about. It is refreshingly free of super-complex Bayesian, likelihoodist, and phylogenetic methods (I have no problem with those approaches, but implementations are often baroque). For example, using generic means instead of creating an overall phylogeny of palms for the sole purpose of estimating fruit sizes in missing species would be overkill. I am also inclined to believe the paper's results, which are intuitive, and the literature treatment seems fair.

However, there has been some methodological shortcutting, and treating the data differently may well alter the results. I specifically suggest adding more analyses that will (1) include all variables in the ordination used to reduce collinearity, (2) use gridded cell data, (3) use a narrower definition of frugivory in extinct species, and (4) explore different sets of future extinction probabilities.

(1) Model averaging (first mentioned on line 100) is a little weird because it treats the universe of potential explanations as consisting of many models of which all but one omit one or more explanatory variables, as if the real world featured 100% orthogonal relationships. Of course it does not, so the undertaking makes no sense. On the other hand, simpler alternatives like stepwise regression also fail because of collinearity, and only using the master "kitchen sink" regression model might obscure some relationships because of collinearity.

A better approach is to reduce collinearity through multivariate ordination and then use ordination scores instead of raw variables. This is actually done for the climate variables (line 108), so why stop there? Why not include the body size and LGM variables, and thereby quantify the collinearities involving them? At a minimum, some reason for instead using model averaging should be given. I would prefer to also see results based entirely on ordination scores. Incidentally, I would prefer to see factor analysis used instead of PCA (see below).

(2) I understand that the "botanical country" data from the world checklist of palms were used instead of, say, GBIF data because the checklist is "curated" (line 309). I very much appreciate the value of curated data. Still, though, the spatial resolution used here is so crude that I really do wonder whether the results would have been different if (say) GBIF data binned into degree cells had been used. I find it hard to believe that using a scheme in which "countries" vary greatly in their area and coverage of

biomes introduces no bias. An additional analysis featuring cell data would therefore be very interesting to see.

(3) Considering all large herbivores to be frugivores (line 370) is a terrible idea. Numerous extinct megafaunal species, such as the million and one species of *Equus* and several extinct species of antilocaprids, to cite just a few, were probably not frugivores at all. Even within groups such as rhinos, some species eat hardly any fruit and some eat a lot. Surely, then, more work could and should have been put into this very crucial part of the analysis.

I realise that the authors will probably not want to change anything about the underlying data, which is too bad. If they don't, then at the very least they could run a separate scenario in which only the most obvious frugivores are identified as such. Tapirs are obvious frugivores, equids aren't, and so on.

This one problem is so fundamental that it seriously calls a key part of the results into question, namely, the finding that Pleistocene frugivore size doesn't predict fruit size as well as current frugivore size. For that reason, I think the authors really shouldn't brush aside the issue with some excuse.

(4) Inferring extinction probabilities from IUCN threat statuses (lines 150 - 153) is a totally non-trivial problem. Personally, I think any such inference would be pretty much pure speculation. It doesn't matter that you can get the numbers from a published paper (Davis et al.) or that two of the current authors were involved in that paper; it matters whether the entire undertaking makes sense, which it does not. At a minimum, something exact could be said about how the probabilities are inferred (I realise that it makes sense to give the actual probabilities only on lines 449 and 450). Better, a couple of other scenarios with different probability values (higher or lower) could also be run.

I have a number of minor comments.

The introduction (six dense paragraphs) is long and details points that are sometimes obvious (e.g., that frugivores and fruits have to have correlated sizes, or that defaunation is a bad thing for large-fruited trees). The main point isn't reached until the fifth paragraph. So I wonder if some of this material could be compacted or moved to the discussion, which itself is very long.

Lines 11 - 13: This sentence is hard to understand because it's not clear what is meant by "magnitude" or "keep up with". Perhaps it could be cut into two sentences and rewritten for clarity.

Lines 88 - 92: The spatial resolution of this study is quite crude, as indicated here. Good place to mention that gridded data show the same thing, if they do.

Lines 93 - 96: The argument for using maxima instead of medians isn't fully convincing. More about this matter is brought up later, but an up-front statement that medians aren't as informative because birds also eat small fruits would be good to include here.

Line 100: Model averaging can involve either Bayesian or likelihoodist approaches. MuMIn is a likelihoodist package, but can implement the BIC. At a minimum, the authors should specify here whether they mean averaging based on the AICc, the BIC, or whatever.

Lines 102 - 103: I don't see an obvious reason not to standardize the coefficients. Why not just use the standardized versions alone?

Line 104: Spatial autocorrelation is a major problem, and arguably the major problem, with the statistical setup of this paper. Dismissing it without giving details is a little frustrating. Surely

something could be said here about the interesting method that was used, which is given full treatment later on but is not explained at all here.

Line 119: It would help to state the global number of extinct "frugivore" species here (a figure for the Neotropics is given below, and the grand total is given much later). Again, more than one "frugivore" definition should be given.

Line 126: The definition of "megafaunal" is very strict. People used to talk about 44 kg as a reasonable cutoff. Why use this very high cutoff?

Line 127: Proboscideans is misspelt.

Line 128: What about Sumatran rhinos?

Line 129: Loxodonta is misspelt.

Lines 134 and 135: This is confusing because the exact same R2 values are repeated in two consecutive sentences. Is there a duplicate sentence, or are the numbers wrong, or is it a big coincidence?

Line 163: 0.53 cm compared with what? Using a percentage instead, or at least in addition, would be more intuitive.

Line 165: I think you mean southeast Asia, not South East Asia.

Line 171: Sizes, not size.

Line 180: 2 cm, not 2cm.

Section starting at line 195: The obvious explanation is that there have been hundreds generations of palms since the megafaunal extinctions in most places, plenty of time for major range shifts to have occurred. This could be pointed out somewhere.

Line 215: This sentence seems to suggest that the current manuscript has been scooped by the Doughty et al. paper. If that's not true, the sentence should be rewritten to explain why that paper's results and conclusions are different.

Line 234: "are more" should be "is more".

Line 263: The 0.53 cm figure comes up here again.

Lines 266 and 269: South-east vs. southeast again.

Line 286: The Mascarenes, not Mascarenes.

Line 290: "decrease", not "decreased".

Line 300: "ameliorated", not "ameriolated".

Line 316: Strike "the" before Indo-Australia.

Line 321: Mammalian frugivores are absent in how many botanical countries? A lot or a few?

Line 345: I think it's a shame that Paul Martin (1967, 1984) isn't credited here or anywhere for making obvious points about Quaternary extinctions such as the correlation between human arrival and extinction. Literature cited here just repeats his arguments.

Line 356: Whose occurrence data? FAUNMAP? Neotoma, which includes FAUNMAP? NOW? PaleoDB? What was used for regions that are not well-represented in those databases? Be specific.

Lines 356 - 357: Estimating occurrence based on co-occurrence is a very dodgy business. It's too bad that this was done. Best to at least explain here exactly how it works.

Line 363: The IUCN prefers being cited explicitly, as opposed to only giving the download date.

Line 383: What does "for each botanical country" mean? An average value across all 30 arc second cells included in each country, or something else?

Line 389: I would prefer a factor analysis to a principal components analysis because the former method cleanly separates groups of variables, whereas the latter often identifies variables as loading strongly on multiple axes. That makes FA trivial to understand and PCA a real mess. FA is super-easy to implement (function `factanal` in the built-in R package `stats`, or `fa` in package `psych`, which offers some advantages). Please explain why PCA was preferred to FA.

Lines 416 to 419: This is a very interesting approach! It's too bad that it was made necessary by the use of botanical countries instead of grid cells (see above).

Lines 424 - 425: No reason is given here for using model averages, and there is no explanation of how they are computed.

Lines 430 - 431: Likewise, MuMIn is cited but functions and parameters are not described.

Line 439: There's a typo ("autocorrelatioon").

Figure 1: The coloured points are a little hard to see against the dark grey country colour. I suggest using a lighter grey.

Figure 2: There is a typo in the caption ("coefficeints"). Green vs. yellow is not the best colour scheme for colour vision deficient people. Blue vs. yellow would be easier to see.

John Alroy

Reviewer comments

Reviewer #1 (Remarks to the Author):

I enjoyed this a lot. It is an important question, probably the best data available to answer it, and the analysis appears to be robust and defensible. I have only a few minor comments and suggestions. First, although the focus on maximum/maximum neatly deals with the problems of palm fruit and frugivore diversity, I think readers need a brief introduction to at least the diversity of palm fruits, which vary greatly in size, external protection, color, and dispersal agents.

We have now added a brief summary of the phenotypic variation in palm fruits to the introduction.

In the defaunated sites in Asia, the largest avian frugivores - fruit pigeons and hornbills - are taking palm fruits (< 4 cm diameter) as large as the largest surviving mammals. I also think you need to be clearer on the consequences of looking at fruit sizes rather than seed size, with the former less relevant for mammals but more easy to get. I also think that the differences between regions were inadequately discussed. It is not obvious to me why regional results should differ at all.

The reviewer is certainly right here. We have now added a sentence in the Discussion acknowledging the important role of large birds such as hornbills and parrots in the dispersal of large palm fruits.

Regarding seed vs. fruit sizes, it is true that fruit sizes do not always approximate seed sizes (e.g. if a fruit has many seeds), but in the case of palms most species are one-seeded. Thus, fruit sizes are highly correlated to seed sizes. We have now added an explanation in the Discussion, and a brief summary of palm fruits in the Introduction to make this clearer. We have further added a bit of text in the discussion when discussing the magnitude of estimated fruit size change.

Regarding regional differences, we have now added a discussion of regional variation in the fruit size – body size relationship in the Discussion, in particular, some possible explanations for the lack of a clear relationship between body size and fruit size in the Afrotropics, in contrast with other biogeographic regions.

Reviewer #2 (Remarks to the Author):

Lim et al. (NCOMMS-19-32434-T) is an interesting and well-written study that will draw more attention to the current extinction crisis and to Quaternary extinctions, which are still not very widely known about. It is refreshingly free of super-complex Bayesian, likelihoodist, and phylogenetic methods (I have no problem with those approaches, but implementations are often baroque). For example, using generic means instead of creating an overall phylogeny of palms for the sole purpose of estimating fruit sizes in missing species would be overkill. I am also inclined to believe the paper's results, which are intuitive, and the literature treatment seems fair.

However, there has been some methodological shortcutting, and treating the data differently may well alter the results. I specifically suggest adding more analyses that will (1) include all variables in the ordination used to reduce collinearity, (2) use gridded cell data, (3) use a narrower definition of frugivory in extinct species, and (4) explore different sets of future extinction probabilities.

We thank the reviewer for the positive feedback and very detailed reading of your manuscript. Below we address each of the reviewer's concerns individually.

Model averaging (first mentioned on line 100) is a little weird because it treats the universe of potential explanations as consisting of many models of which all but one omit one or more explanatory variables, as if the real world featured 100% orthogonal relationships. Of course it does not, so the undertaking makes no sense. On the other hand, simpler alternatives like

stepwise regression also fail because of collinearity, and only using the master "kitchen sink" regression model might obscure some relationships because of collinearity.

We believe model-averaging is appropriate in our specific case because the true model is rarely known nor is the true model necessarily within the set of candidate models. Weighting the effect size of variables in different models based on their relative support (i.e., Akaike weights) provides more reliable estimates of effect sizes than stepwise deletion of variables. We have now provided further justification and more details on how the model averaging procedure was implemented in the Results and Methods.

On the issue of collinearity in the full models (with all variables included), collinearity was low among predictor variables (variance inflation factors for all variables were < 4), and thus did not have a large impact on model averaged effect sizes. Nonetheless, we additionally performed an alternate model-averaging approach where coefficients were standardized based on their partial standard deviations prior to averaging (following Cade 2015). This approach provides effect sizes for each variable that take into account its collinearity and explanatory power of other variables. The results do not appear to vary strongly between these two approaches, probably because of the low collinearity between variables.

A better approach is to reduce collinearity through multivariate ordination and then use ordination scores instead of raw variables. This is actually done for the climate variables (line 108), so why stop there? Why not include the body size and LGM variables, and thereby quantify the collinearities involving them? At a minimum, some reason for instead using model averaging should be given. I would prefer to also see results based entirely on ordination scores. Incidentally, I would prefer to see factor analysis used instead of PCA (see below).

We agree that collinearity can be an issue in multiple regressions. As mentioned above, in our case collinearity was low (variance inflation factors for all variables were < 4). Additionally, we have performed an alternate model-averaging approach which takes into account collinearity between variables. The results from this approach did not vary significantly from our default analysis. With regards to the ordination of variables, the reason why we only ordinated climatic variables was to maintain interpretability of body size relative to other variables (see more details below).

(2) I understand that the "botanical country" data from the world checklist of palms were used instead of, say, GBIF data because the checklist is "curated" (line 309). I very much appreciate the value of curated data. Still, though, the spatial resolution used here is so crude that I really do wonder whether the results would have been different if (say) GBIF data binned into degree cells had been used. I find it hard to believe that using a scheme in which "countries" vary greatly in their area and coverage of biomes introduces no bias. An additional analysis featuring cell data would therefore be very interesting to see.

We used the world checklist of palms (with 'botanical countries' as standardized sampling units as defined by the International Working Group on Taxonomic Databases, TDWG) not only because the data are curated, but also because it is the only available global dataset that provides quality-checked presence-absence data for all (>2500) species of palms worldwide. While $>900,000$ observations (i.e. presence-only records) of palms are available in the GBIF database, the records are highly biased both geographically and taxonomically. For instance, nearly 30% of all palm species have not even a single occurrence record in GBIF, and only 60% have more than 1 record. But even for the latter species, occurrence records are strongly biased in geographic space because field sampling is limited and commonly restricted to easily accessible sites (national parks, close roads or cities etc.). Furthermore, many records do not even have geo-referenced coordinates, or may have errors in the provided latitude-longitude coordinates. Hence, using GBIF occurrence records for a global analysis at the level of whole species assemblages (e.g. by binning them into grid cells) is highly problematic because it will result in flawed analysis due to the inherent biases of GBIF records. A recent high-profile study (Meyer et al. 2016, *Ecology Letters* 19, 992–1006) has quantified this at a global scale using GBIF records for selected plant families (incl. palms) and shows the massive gaps and problems associated with the multidimensional data

limitations of GBIF records for assemblage-level analyses (opposed to focusing on single species with a good coverage). Similar results have been shown for GBIF records of animals including mammals (Meyer et al. 2015, Nature Communications 6, 8221).

In contrast to some terrestrial vertebrate groups (e.g. birds, mammals, amphibians and reptiles), range maps for all species of palms (and in fact for all species-rich, tropical plant families) do not exist globally. Moreover, GBIF observations are presence-only records and do not provide information on absences, but this is crucial for assemblage-level analyses because otherwise the results will be biased due to false absences. Hence, the only possibility to conduct a reliable global analysis of palms (and any other species-rich, tropical plant family) is to use the quality-checked presence-absence data from the World Checklist of Palms. While we agree that the spatial resolution of 'botanical countries' is coarse, it is currently the best scale to conduct a meaningful and reliable, global analysis at the level of species assemblages. An additional reason for using data from the World Checklist of Palms is that the taxonomy (curated by leading palm taxonomists) has been harmonized so that recently compiled trait data can seamlessly be integrated (see Scientific Data, 6, 178).

We therefore truly believe that this dataset represents the most comprehensive and most accurate representation of palm assemblages worldwide. To better explain this in the manuscript, we have added text and additional justification in the methods. We further included reference to Meyer et al. (2016) to refer to the large gaps and uncertainties in global occurrence information of plants.

(3) Considering all large herbivores to be frugivores (line 370) is a terrible idea. Numerous extinct megafaunal species, such as the million and one species of *Equus* and several extinct species of antilocaprids, to cite just a few, were probably not frugivores at all. Even within groups such as rhinos, some species eat hardly any fruit and some eat a lot. Surely, then, more work could and should have been put into this very crucial part of the analysis.

I realise that the authors will probably not want to change anything about the underlying data, which is too bad. If they don't, then at the very least they could run a separate scenario in which only the most obvious frugivores are identified as such. Tapirs are obvious frugivores, equids aren't, and so on.

This one problem is so fundamental that it seriously calls a key part of the results into question, namely, the finding that Pleistocene frugivore size doesn't predict fruit size as well as current frugivore size. For that reason, I think the authors really shouldn't brush aside the issue with some excuse.

This is a really good point and we appreciate your input. Following your suggestion, we re-analyzed the data with two nested and more conservative frugivore classifications to address uncertainty in such classifications. Our "Very conservative" classification considers only extinct taxa in extant families that are predominantly ($\geq 50\%$) frugivorous (based on classifications in the MammalDiet dataset), but we included a less strict definition that makes exceptions for groups for which modern analogs might be poor proxies (e.g., sloths), taxa that are borderline cases ($\geq 40\%$ of modern taxa in the family frugivores), or taxa that appear to be mostly browsers or mixed-feeders. A summary of these new conservative definitions has now been included in the Methods of the main text, with more details (i.e., specific references supporting the inferred diet of extinct taxa and table listing the classification of extinct mammals under the different definitions) in the Supplement.

Our main results under these conservative frugivore classifications were not significantly different from those under our default classification. Current mammalian frugivore assemblages better explain present-day palm patterns compared to present-nature frugivore assemblages except in the Neotropics. This pattern in S. America, however, was not borne out with the most conservative circumscription. This was likely because many Pleistocene S. America frugivorous taxa (e.g., mega-sloths) do not have particularly frugivorous extant relatives, and hence were excluded in the most conservative definition.

(4) Inferring extinction probabilities from IUCN threat statuses (lines 150 - 153) is a totally non-trivial problem. Personally, I think any such inference would be pretty much pure speculation. It doesn't matter that you can get the numbers from a published paper (Davis *et al.*) or that two of the current authors were involved in that paper; it matters whether the entire undertaking makes sense, which it does not. At a minimum, something exact could be said about how the probabilities are inferred (I realise that it makes sense to give the actual probabilities only on lines 449 and 450). Better, a couple of other scenarios with different probability values (higher or lower) could also be run.

This is a fantastic idea and we agree that we might have overly relied on the original papers. The papers by Mooers *et al.* (2011, PLoS ONE) and Davis *et al.* (2018; PNAS) partially used the Red List designations / criteria as rough guidelines. For example, the designation of a species that was critically endangered (CR) was one that had a 1 in 2 chance of going extinct in 10 years; endangered (EN) species was 1 in 5 in 20 years and a vulnerable (VU) species had a 1 in 10 chance of going extinct in 100 years (see IUCN Red List Categories and Criteria Version 3.1). Assuming extinction were constant rate processes in each category, they fixed the probability of a Least Concern (LC) species at 0.01% in 100 years and then assumed the Near Threatened (NT) category had a probability of extinction around 100 times this based on an interpolation. The study by Davis *et al.* (2018), as you rightly identified as having common authors with this manuscript, used a different methodology to derive their own rates for the NT and LC categories by extrapolating extinction rates for the threatened categories (CR, EN and VU) and assuming an exponential decrease in extinction rate across red list categories (fitted a linear regression with red list status as an ordinal variable; the R² of this regression was 0.99). They additionally validated the extinction probabilities of LC species with a dataset by Di Marco *et al.* (2014; more details below) and derived a probability of extinction of 0.4% for LC species over 100 years, which was higher than their own probabilities (=0.17%) by two-fold which suggests that their rates for LC species are still conservative.

Here, to address your concerns, we additionally derive extinction probabilities using two datasets that we think provides an additional set of rates that we then use as an alternate, conservative extinction scenario. These are datasets (Hoffmann *et al.* 2011 *Science* and Di Marco *et al.* 2014 *Conservation Biology*) that have tracked changes in Red List categories for all mammals (Hoffmann) or ungulates + carnivorans (Di Marco), over 12- and 33-year periods respectively. Red List category changes (and lack thereof) were validated on a case-by-case basis by the authors. This is crucial as Red List criteria have changed somewhat over the years and so changes in status may be artificial.

We then estimated the transition rates between IUCN Red List categories from these two datasets by fitting a continuous-time Markov chain model using maximum likelihood (ML) (more details in the Supplement). This approach is novel and allowed us to calculate the extinction probabilities for a species in any Red List category for any arbitrary time frame. These probabilities are of course dependent upon the estimated rates based on transition rate changes before and that the rates are constant through time.

To estimate the reliability of these ML rate estimates, we used a parametric bootstrap approach to simulate data given the same starting distribution of species across Red List categories for each of the two studies, and the same amount of time representing their respective observational time frames. We then fit the same model to these simulated data and compared our empirical estimates to estimates derived from these simulations. ML-estimated rates appear to be reliable as they fall quite comfortably within the rates obtained across bootstrap simulations.

Extinction probabilities estimated from these datasets for less threatened categories were close to those used in Davis *et al.* (2018) although the extinction probabilities for threatened categories (VU, EN and CR) are far lower than used in those studies (Table S9 of the Supplementary Information; also provided below). This is perhaps unsurprising given the data: out of the 138 CR mammals in the Hoffmann *et al.* (2011) dataset, only 4 species went extinct over the 12-year

period evaluated, whereas the Red List status of 5 species actually showed improvement. However, using the rate from Davis *et al.* (2018) and assuming a constant-rate model ($\text{Prob}(\text{survival}) = e^{-0.069 \cdot 12}$), one should expect about half of the 138 species to have gone extinct. This discrepancy is likely due to the context-specific nature of IUCN criteria. For example, the IUCN definition for a CR species is one with an extinction probability of 50% in 10 years, or 3 generations (whichever is longer). This approach would arguably overestimate extinction risk of long-lived species in this approach if IUCN categories are applied universally, though it is also possible that the Red List designations may have been on the pessimistic side.

Table S9: Extinction probabilities of species of different IUCN Red List statuses over a 100 year period. Probabilities inferred using Marco *et al.* 2014 and Hoffmann *et al.* 2010 datasets were derived using maximum likelihood rate estimates of the CTMC model (equation 1, 2). Extinction probabilities of threatened categories from Davis *et al.* 2018 are derived from IUCN Red List criteria (Mooers *et al.*, 2008). Extinction rates for LC and NT categories in Davis *et al.* 2018 were extrapolated from rates for threatened taxa by assuming an exponential change in extinction rate by assuming IUCN categories as an ordinal variable.

IUCN Red List category	Hoffmann et al (2010) (All mammals)	Di Marco et al (2014) (Carnivores + ungulates)	Davis et al (2018) (Red List definitions and extrapolation)
LC	0.0001	0.0026	0.0017
NT	0.0041	0.0199	0.0141
VU	0.0139	0.0385	0.1000
EN	0.0548	0.0844	0.6723
CR	0.1802	0.2048	0.9990

Ultimately, we used the average of the two sets of CTMC-derived rates and the Davis *et al.* (2018) rates. We feel these values represent appropriate upper and lower-bound estimates of extinction probabilities because 1) while the CTMC-derived rates are lower than those of Davis *et al.*, they do not take into account the possibility that anthropogenic impacts (and hence extinction rates) may intensify in future (certainly a plausible scenario!), and 2) while the Davis rates are somewhat high, they at least represent an intrinsic classification of extinction risk defined by the IUCN. True extinction probabilities are thus likely to be somewhere within the range of the two sets of rates. We now also provide an estimate of projected fruit size needed to keep up with changes in frugivore body size change assuming a 100-year time window (instead of a 50 year one in the original submission). Upon reflection, we feel this to be a more salient time window given the time scale of extinction and ecological and evolutionary changes in palm populations (e.g., the Galetti paper 2013 report evolution in seed sizes in the order of hundreds of years since defaunation).

We now also take the average values across simulations instead of the median. This is because four botanical countries (= Rodrigues Isl, Mauritius, Seychelles, Ogasawa Isl.) have so few species of mammalian frugivores (= only fruit bats) that in some of the simulations, they are completely extirpated (body size is now zero). Taking the median is very insensitive to this loss and underrepresents the potential decline of body mass in those botanical countries.

Details on how the new set of extinction probabilities were derived, parametric bootstrapping are now in the main text and the Supplementary Information. All projections of fruit size distributions under future mammal extinction scenarios were now also calculated with these additional rates, and figures and text were updated.

I have a number of minor comments.

The introduction (six dense paragraphs) is long and details points that are sometimes obvious (e.g., that frugivores and fruits have to have correlated sizes, or that defaunation is a bad thing for

large-fruited trees). The main point isn't reached until the fifth paragraph. So I wonder if some of this material could be compacted or moved to the discussion, which itself is very long.

We have now shortened parts of the introduction, especially in the second paragraph. We also reformulated some other parts to make the text smoother.

Lines 11 - 13: This sentence is hard to understand because it's not clear what is meant by "magnitude" or "keep up with". Perhaps it could be cut into two sentences and rewritten for clarity.

The sentence has now been cut into two sentences, and we are now more explicit in our reference to the way these numbers are derived to improve clarity: "Simulations of frugivore extinction over the next 100 years suggest that changes in body size will require a 4 - 9 % assemblage-level decrease in palm fruit sizes to maintain the global frugivore body size - fruit size relationship. Absolute changes in assemblage-level means of palm fruit size were between four- to six-fold higher than published species-level estimates of seed size change following defaunation"

Lines 88 - 92: The spatial resolution of this study is quite crude, as indicated here. Good place to mention that gridded data show the same thing, if they do.

Please see our response to gridded analyses / GBIF occurrences above.

Lines 93 - 96: The argument for using maxima instead of medians isn't fully convincing. More about this matter is brought up later, but an up-front statement that medians aren't as informative because birds also eat small fruits would be good to include here.

A short sentence explaining this point has been added.

Line 100: Model averaging can involve either Bayesian or likelihoodist approaches. MuMIn is a likelihoodist package, but can implement the BIC. At a minimum, the authors should specify here whether they mean averaging based on the AICc, the BIC, or whatever.

You are absolutely right. Model averaging was performed using Akaike weights that were calculated using AICc. These details are now included in the Methods section.

Lines 102 - 103: I don't see an obvious reason not to standardize the coefficients. Why not just use the standardized versions alone?

We would like to clarify that the coefficients are already standardized by first scaling all predictor variables to unit variance and centered. This is a standard procedure to allow for different coefficients of different predictor variables to be more directly comparable. What we are referring here is an additional standardization of the standardized coefficients themselves prior to model-averaging. This approach is not yet widely adopted but has the advantage of taking into account the effect of collinearity between predictor variables on model-averaged coefficients. To avoid potential confusion, we have now improved our description and justification of this model-averaging standardization approach in the Results and Methods.

Line 104: Spatial autocorrelation is a major problem, and arguably the major problem, with the statistical setup of this paper. Dismissing it without giving details is a little frustrating. Surely some thing could be said here about the interesting method that was used, which is given full treatment later on but is not explained at all here.

We have now better introduced the spatial statistical approach in the Results, i.e., where they are first mentioned.

Line 119: It would help to state the global number of extinct "frugivore" species here (a figure for the Neotropics is given below, and the grand total is given much later). Again, more than one "frugivore" definition should be given.

As described above, we now provide two additional classifications (see above). The number of extinct frugivores considered in each of these definitions is now reported in the manuscript at this place in the results.

Line 126: The definition of "megafaunal" is very strict. People used to talk about 44 kg as a reasonable cutoff. Why use this very high cutoff?

This cutoff was for illustrative purposes only and is typically used in the context of herbivores, following Owen-Smith (1988). However, both definitions (> 44 and > 1000) are arbitrary to some degree and the precise cutoff used does not have any bearing on our quantitative analyses per se. We have reworded this sentence to avoid giving the impression that we are actually defining the term: "21 potential megafaunal (> 1,000 kg) mammalian frugivores .." has been changed to "21 potential megafaunal mammalian frugivores with body masses above 1,000 kg ...".

Line 127: Proboscideans is misspelt.

Corrected

Line 128: What about Sumatran rhinos?

Species name for Sumatran rhinos is now included.

Line 129: Loxodonta is misspelt.

Corrected

Lines 134 and 135: This is confusing because the exact same R2 values are repeated in two consecutive sentences. Is there a duplicate sentence, or are the numbers wrong, or is it a big coincidence?

Yes, you are absolutely right and thank you for spotting this error. The numbers referred to the R2 values, the correct coefficients were however in the original Table S1. Numbers in the manuscript are now updated.

Line 163: 0.53 cm compared with what? Using a percentage instead, or at least in addition, would be more intuitive.

This is the difference in predicted fruit size (using simulated future frugivore assemblages) and the current fruit size across botanical countries. The average percentage loss (relative to fitted values) is now provided.

Line 165: I think you mean southeast Asia, not South East Asia.

All instances of South-east Asia or South East Asia are now Southeast Asia (following the prevailing convention)

Line 171: Sizes, not size.

Corrected.

Line 180: 2 cm, not 2cm.

Corrected.

Section starting at line 195: The obvious explanation is that there have been hundreds generations of palms since the megafaunal extinctions in most places, plenty of time for major range shifts to have occurred. This could be pointed out somewhere.

This is a good point and we agree with your point on the generation times of palms. This was already implied in this section, in our argument that the Neotropics have had less time and have had more intense defaunation than other biogeographic realms. We have rewritten that paragraph and added some more emphasis on the time scale.

Line 215: This sentence seems to suggest that the current manuscript has been scooped by the Doughty et al. paper. If that's not true, the sentence should be rewritten to explain why that paper's results and conclusions are different.

The Doughty paper arrives at their conclusion through simulation models. The sentence has been rewritten to better reflect this.

Line 234: "are more" should be "is more".

Corrected.

Line 263: The 0.53 cm figure comes up here again.

Sentence has been corrected.

Lines 266 and 269: South-east vs. southeast again.

Corrected (see above).

Line 286: The Mascarenes, not Mascarenes.

Corrected.

Line 290: "decrease", not "decreased".

Corrected.

Line 300: "ameliorated", not "ameriolated".

Corrected.

Line 316: Strike "the" before Indo-Australia.

Corrected.

Line 321: Mammalian frugivores are absent in how many botanical countries? A lot or a few?

This was only the case for one botanical country, Reunion Island, which has lost all of its mammalian frugivores (two species of fruit bat, *Pteropus niger* and *Pteropus subniger*). There is at present a newly established but very small population of *Pteropus niger* on Reunion Island that has originated from Mauritian stock (probably in the last several years; <https://www.iucnredlist.org/species/18743/86475525>), but this is a fairly recent event (compared to when it originally went extinct on Reunion: between 1772 and 1801) and for consistency we have decided not to update its status. A natural history reference has been added supporting the timing of their putative extinction.

Line 345: I think it's a shame that Paul Martin (1967, 1984) isn't credited here or anywhere for making obvious points about Quaternary extinctions such as the correlation between human arrival and extinction. Literature cited here just repeats his arguments.

Thank you for pointing out this omission. We have now included these references, and for balance, two recent studies that suggest that climate change may have played an interacting role with anthropogenic factors.

Line 356: Whose occurrence data? FAUNMAP? Neotoma, which includes FAUNMAP? NOW? PaleoDB? What was used for regions that are not well-represented in those databases? Be specific.

Occurrence data came in large part from FAUNMAP (for N. America) and the Paleobiology database (for remaining areas). Primary literature was used to supplement these databases for taxa outside of the New World for species with insufficient records. References are in the original papers describing the dataset (Faurby and Svenning 2015; Faurby et al 2018). This has been added to the Methods section.

Lines 356 - 357: Estimating occurrence based on co-occurrence is a very dodgy business. It's too bad that this was done. Best to at least explain here exactly how it works.

This was performed by identifying grid cells that contained at least 50% of extant taxa that co-occurred with an extinct taxon in fossil assemblages. More details on this methodology behind the range maps of extinct taxa in the PHYLACINE database (Faurby et al 2018) are now reported in the Methods.

Line 363: The IUCN prefers being cited explicitly, as opposed to only giving the download date.

This is now done. We now also cite the Red List criteria in reference to the extinction risk probabilities used by Mooers *et al.* 2011 and Davis *et al.* 2018.

Line 383: What does "for each botanical country" mean? An average value across all 30 arc second cells included in each country, or something else?

Yes, that is correct. Additional clarification is now provided.

Line 389: I would prefer a factor analysis to a principal components analysis because the former method cleanly separates groups of variables, whereas the latter often identifies variables as loading strongly on multiple axes. That makes FA trivial to understand and PCA a real mess. FA is super-easy to implement (function `factanal` in the built-in R package `stats`, or `fa` in package `psych`, which offers some advantages). Please explain why PCA was preferred to FA.

In our opinion, a factor analysis would reduce the interpretability of the variables, especially the core predictor variable that we are interested in (i.e., body size). We have thus not adopted this approach. PCA was only performed on the present-day climatic variables as we were not interested in the precise variables themselves and they were collinear with each other but not with the others (as evidenced in the low variance inflation factors). We added a sentence in the methods to better explain this.

Lines 416 to 419: This is a very interesting approach! It's too bad that it was made necessary by the use of botanical countries instead of grid cells (see above).

Yes, we are glad you liked our solution. We have provided further justification for our use of botanical country as opposed to grid cells in the manuscript.

Lines 424 - 425: No reason is given here for using model averages, and there is no explanation of how they are computed.

We have now provided more details on how the averaging was performed in the Methods, with additional references.

Lines 430 - 431: Likewise, MuMIn is cited but functions and parameters are not described.

The functions used, and parameters (for the standardization routine described by Cade 2015) are now reported. We would also like to add that the code / wrapper functions that we used are already openly available (see <https://github.com/junyinglim/megafaunalFrugivore>), so readers may also use that as reference. If published, our scripts will also be archived.

Line 439: There's a typo ("autocorrelat~~io~~on").

Corrected.

Figure 1: The coloured points are a little hard to see against the dark grey country colour. I suggest using a lighter grey.

Done.

Figure 2: There is a typo in the caption ("coefficeints"). Green vs. yellow is not the best colour scheme for colour vision deficient people. Blue vs. yellow would be easier to see.

Thank you. Typo corrected, and colour-scheme has been changed.

John Alroy

Reviewers' Comments:

Reviewer #1:

Remarks to the Author:

This is an excellent revision and I have no additional suggestions.

Reviewer #3:

Remarks to the Author:

Overall:

Lim et al. perform some macroecological analysis to study the relationship between mammal body size / palm tree fruit size. They find a correlation between maximum fruit size and mammal body size in most regions of the world. They also explore this same relationship including Late Pleistocene mammals in order to test whether palm trees are still adapted to their Pre-megafaunal extinction fruit dispersers. They found that current fruit size is better explained by modern faunas alone. The project is really interesting and the results worth communicating to a broad audience. The authors do a particularly good job with the figures. However, I think the text needs some proof-reading and some sentences need clarification. I outline further comments below.

Additionally, I reviewed author's response to Reviewer2's comments at the end of my own comments..

Some paragraphs need more citations and an explanation of the reasoning behind the causes of observed relationships (see some examples below).

What is the reason for analyzing Asia and Australia together? Australia suffered a dramatic megafaunal extinction, and the biggest mammal in the continent is many orders of magnitude smaller than the biggest mammal in Asia. How do you think this could affect your results?

I understand that this is a paper about mammal body size in relationship to fruit size in palms. However, other seed dispersing species might be playing a big role on this relationship. While mentioned in the discussion, I think that more attention should be given to this, perhaps a whole paragraph evaluating a few consequences on your results.

Additionally, what is the effect of plant domestication on your analysis? Are you accounting for the fact that some palm species might have bigger fruits as a result of human domestication? Some justification on the matter would be needed.

Abstract

First sentence (lines 2-3): I would say "influences the structure and composition of plant communities", but this is just a suggestion.

Line 17-18: I suggest: "the impact of the extinction of seed dispersers".

Introduction

Line 22-24: I find it hard to follow what you mean with this sentence

Throughout: Try to avoid repeating the same words. Just the first paragraph says "seed dispersal" 4 times, in 7 lines of text.

Explain "ecological anachronism", not all readers are ecologists.

Methods:

The authors did a good job explaining the different steps in their analysis and the justifications for

their criteria.

Minor comment: They specify "These values are potentially conservative, as they assume rates of transition between Red List criteria to be constant through time, whereas anthropogenic impacts may intensify into the future". I disagree. To my understanding, if a species has the potential of being more threatened in the future and you are considering a "less" threatened state (modern), you are being less conservative in your analysis. You'd be more conservative if you established which species are more likely to increase their threatened state and applied it to the analysis. But since this is a matter of opinion and/or perspective, I'd remove the statement about being conservative.

More information should be provided about the extinction simulation. Additionally, I think it would be sensitive to incorporate the provability of a plant species going extinct as well. I understand that these would make this project even more complex. But since many readers might wonder about why this wasn't considered, I would add a few sentences on why it wasn't incorporated in this study.

Results:

This section is well articulated and does a good job summarizing methods before proceeding with the numbers.

Lines 180-182: add "see Methods", or specify what predictors as it is unclear what you mean with "all other predictors".

Discussion

Lines 207-211: You should explain the reasons for not observing a relationship between frugivore body size and fruit size in Afrotropics a bit more. Right now your train of thought or reasoning behind is not clear from the text.

Line 229: Cite

I think a more extended evaluation of the role of birds in fruit dispersal and how not including them in the study could affect your results. Do you think they could be responsible for dispersing the smaller-to-mid-sized fruits and therefore have something to do with the lower relationship between fruit size and mammal body size? I don't think a single sentence covered the whole ecological and historical effects they might be having in the studied relationship.

Lines 244-247: A few citations are needed for both statements. Look at the literature covering evolutionary responses and timing of fruiting plants.

Line 273-274: Can you justify this claim with a citation: "whereas the dispersal of the largest palm fruits may only be effectively performed by large mammals"?

Line 292-294: I find this sentence confusing. Are you saying that it was more likely for plants to go extinct than adapt (evolve smaller fruits)?

Line 314: You can remove the "thus" since you already used "since" earlier in the sentence.

Conclusions

Lines 343-346. This is a really long somehow confusing sentence. It also may need citations

Figures

Figures are really well made, neat and clear. It is also appreciated that they chose a color-blind friendly scheme.

Figure 1: Define present-natural in figure caption as well

Comments from response to reviewers

This is my take on the answers to Reviewer2's concerns.

Model averaging issue

I'm not up-to-date with model-averaging literature, but I don't think they do a good job at justifying its use. With so much literature on the topic, a better explanation should be provided beyond Reviewer2's concerns. Additionally, when they mention "fitting a set of candidate models", I can't find what models are in this "set". So overall, I think this section needs more work. I'm sure they can find literature to justify the use of model averaging and keep analysis as they are.

Use of ordination analyses

I agree with Reviewer2 that ordinating a subset of variables only might not be the best approach. Alternatively, I propose two fast-achieving analysis. One in which you ordinate ALL variables together as proposed by Reviewer 2 to test whether results are comparable. Another in which you do NOT ordinate any of the variables and apply model-averaging to all variables, including each of the 6 climate variables. This does not need to be in the main manuscript. Just mention it in the text and include it in supplementary materials. It is likely that more readers come up with the same issue.

Gbif data

I understand the author's concerns about GBIF data, but I also agree it would be informative to see the same analysis being performed with GBIF data. I don't think they need to do the analysis, but include more justification in the text/supplements. They do a really good job in the response, use it.

Frugivores

Reviewer 2 raises an important point, something I would have been concerned about as well from the earlier version of the manuscript. I like that they repeated the analysis with what they call a "more conservative" definition. I'm not sure I would describe the former as a "conservative" though. When it comes to fossil species, a simple literature review could shed some light on whether considering some of the mammals frugivores or not. For instance, we know about giant sloth diet from their coprolites. If you treated differently animals for which modern analogs would be poor proxies, you could potentially do a literature search for them. This would make the justification for including them stronger.

Extinction risk

I think they did a good job addressing concerns.

Megafaunal cutoff

I am not concerned about them using the 1,000Kg cutoff, but I'd appreciate a justification in the text if they are not using the traditional 44Kg. Choosing the number was not as arbitrary as they make it sound. There is a large amount on paleontological research on the matter.

Silvia Pineda-Munoz

Reviewer comments

Reviewer #1 (Remarks to the Author):

This is an excellent revision and I have no additional suggestions.

Thank you. We appreciate your comments and suggestions

Reviewer #3 (Remarks to the Author):

Overall:

Lim et al. perform some macroecological analysis to study the relationship between mammal body size / palm tree fruit size. They find a correlation between maximum fruit size and mammal body size in most regions of the world. They also explore this same relationship including Late Pleistocene mammals in order to test whether palm trees are still adapted to their Pre-megafaunal extinction fruit dispersers. They found that current fruit size is better explained by modern faunas alone.

The project is really interesting and the results worth communicating to a broad audience. The authors do a particularly good job with the figures. However, I think the text needs some proof-reading and some sentences need clarification. I outline further comments below.

Additionally, I reviewed author's response to Reviewer2's comments at the end of my own comments..

Some paragraphs need more citations and an explanation of the reasoning behind the causes of observed relationships (see some examples below).

Please see below for a more specific, line-by-line description of the changes we made in response to your comments.

What is the reason for analyzing Asia and Australia together? Australia suffered a dramatic megafaunal extinction, and the biggest mammal in the continent is many orders of magnitude smaller than the biggest mammal in Asia. How do you think this could affect your results?

Add this to the appendix as a sensitivity analysis.

Thank you for your valid point. The main reason why we analysed Asia and Australia together was because we aimed for three broad regional analyses (besides the global one) to capture major biogeographic splits such as between the New and Old World and between Old World East and Old World West. As you correctly mentioned, megafaunal extinctions in Australia (and other parts of the Sahul shelf) were quite drastic compared to the rest of Asia and the very largest extant Australian mammals are much smaller than the largest mammals in Asia. We have now addressed this by including a sensitivity analysis in the supplement where "botanical countries" (BCs) of Australia and New Guinea (n = 6), i.e., the Sahul shelf, were excluded (Supplementary Figure 2). We could not do a separate analysis for only the Sahul shelf because of the small sample size.

We find that present-natural body size showed a stronger relationship with current fruit size, compared with current frugivore assemblages. When Australia + New Guinea was removed, the results for the Indotropics are qualitatively similar so the difference in present-natural vs. current scenarios does not appear to be driven just by the severity of megafaunal extinctions across the Sahul Shelf. We have provided the justification for this sensitivity analysis in the Methods and our findings in the Results.

I understand that this is a paper about mammal body size in relationship to fruit size in palms. However, other seed dispersing species might be playing a big role on this relationship. While mentioned in the discussion, I think that more attention should be given to this, perhaps a whole paragraph evaluating a few consequences on your results.

We have now reworked the first few paragraphs of our discussion to better explain why mammalian frugivores (as opposed to other seed dispersal guilds) are likely to be dominant key drivers of this relationship, namely because mammalian frugivores tend to disperse larger palm fruits than birds, and because they are overall larger and not as limited by gape size compared with non-mammalian (e.g., avian) dispersers. We also now discuss some exceptions for where we expect this pattern to be less strong, for example, in areas where maximum fruit size is small enough (e.g., oceanic islands) that it is unlikely for mammals alone to be shaping fruit size variation.

Additionally, what is the effect of plant domestication on your analysis? Are you accounting for the fact that some palm species might have bigger fruits as a result of human domestication? Some justification on the matter would be needed.

No, human domestication is not relevant here. All trait values were obtained from wild populations (e.g. from taxonomic revisions or ecological field studies) and thus human cultivation does not affect the assemblage-level estimates of maximum fruit size that we use. We have now added to the methods that trait values come from wild specimens.

Abstract

First sentence (lines 2-3): I would say “influences the structure and composition of plant communities”, but this is just a suggestion.

Agreed. This has been amended.

Line 17-18: I suggest: “the impact of the extinction of seed dispersers”.

Thank you. This has been changed.

Introduction

Line 22-24: I find it hard to follow what you mean with this sentence

Throughout: Try to avoid repeating the same words. Just the first paragraph says “seed dispersal” 4 times, in 7 lines of text.

The first paragraph and relevant parts of the introduction have been reworked to reduce repetition.

Explain “ecological anachronism”, not all readers are ecologists.

The original sentences -- “On the one hand, some species with megafauna-adapted fruits may persist as ecological anachronisms. Such fruits may have evolved for dispersal by now-extinct frugivores but are now relatively ill suited for dispersal by remaining present-day frugivores.” -- have been rewritten to emphasize the link between the term and the definition: “On the one hand, some species with megafauna-adapted fruits may persist as apparent “ecological anachronisms” which means that fruits may have evolved for dispersal by now-extinct frugivores but are now relatively ill suited for dispersal by remaining present-day frugivores.”

Methods:

The authors did a good job explaining the different steps in their analysis and the justifications for their criteria.

Minor comment: They specify “These values are potentially conservative, as they assume rates of transition between Red List criteria to be constant through time, whereas anthropogenic impacts may intensify into the future”. I disagree. To my understanding, if a species has the potential of being more threatened in the future and you are considering a “less” threatened state (modern), you are being less conservative in your analysis. You’d be more conservative if you established which species are more likely to increase their threatened state and applied it to the analysis. But since this is a matter of opinion and/or perspective, I’d remove the statement about being conservative.

Thank you for explaining your perspective. Our main argument is that the extinction rates and associated probabilities for our “lower” bound are probably too low because they are estimated on Red List changes in the past. We have changed the language in response to your comments.

The sentence now reads: “These values may give rise to potentially low estimates of extinction risk, as they are based on past changes in Red List status, whereas anthropogenic impacts may intensify into the future”.

More information should be provided about the extinction simulation. Additionally, I think it would be sensitive to incorporate the probability of a plant species going extinct as well. I understand that these would make this project even more complex. But since many readers might wonder about why this wasn’t considered, I would add a few sentences on why it wasn’t incorporated in this study.

Most plant extinctions take a long time and modern extinction risk is additionally dependent upon other stressors that may be unrelated to frugivore / seed disperser loss (e.g., land use change, overharvesting, habitat modification). We have now added additional discussion and references relevant to the reviewer’s point (Line 364 - 371).

Results:

This section is well articulated and does a good job summarizing methods before proceeding with the numbers.

Lines 180-182: add “see Methods”, or specify what predictors as it is unclear what you mean with “all other predictors”.

Done. We now refer readers to the methods,

“Maximum (95th percentile) body sizes of future frugivore assemblages (mean value across 1000 simulations) were then used to generate predictions of future maximum (95th percentile) fruit size assuming the current maximum fruit size – body size relationship (Methods).”

Discussion

Lines 207-211: You should explain the reasons for not observing a relationship between frugivore body size and fruit size in Afrotropics a bit more. Right now your train of thought or reasoning behind is not clear from the text.

We have now re-written that paragraph to discuss and evaluate some potential explanations (e.g., other frugivorous groups or climatic filters on fruit size) which may explain the decoupling between biogeographic variation in body size and fruit size. We also added that this deserves greater study, e.g. to what extent it holds for other fleshy fruited species or how climate may shape the biogeographic distribution of maximum fruit sizes.

Line 229: Cite

I think a more extended evaluation of the role of birds in fruit dispersal and how not including them in the study could affect your results. Do you think they could be responsible for dispersing the smaller-to-mid-sized fruits and therefore have something to do with the lower relationship between fruit size and mammal body size? I don't think a single sentence covered the whole ecological and historical effects they might be having in the studied relationship.

We agree that birds play an important role in the dispersal of small-to-mid-sized fruits, which is probably why the relationship between median mammalian frugivore body size and median fruit size is less strong. We have now reworked the first two paragraphs of the discussion to better explain why biogeographic variation in maximum fruit size is mainly shaped by interactions between mammals and birds. In response to the reviewer's suggestion of historical effects, we now also mention where we expect this relationship to be less clear, for example, in areas where maximum fruit size is small enough that it is unlikely for mammals alone to be shaping fruit size variation (e.g., oceanic islands).

Lines 244-247: A few citations are needed for both statements. Look at the literature covering evolutionary responses and timing of fruiting plants.

Thank you. A couple of citations on the chronology of megafaunal extinctions as well as the generation times of palms have been added for context on the temporal scale of defaunation. The sentence now reads:

“Even when considering that some species of palm may live up to hundreds of years old (Dransfield et al 2008), the time scale of megafaunal extinction (Barnosky et al 2004, Sandom et al 2014) still allows for potentially hundreds or thousands of generations for palm populations to respond to defaunation.”

Line 273-274: Can you justify this claim with a citation: “whereas the dispersal of the largest palm fruits may only be effectively performed by large mammals”?

We have now added two citations to support this statement: Zona and Henderson (1999) and Munoz et al 2019 (J Biogeog). The former is a comprehensive review of animal-mediated dispersal in palms, whereas the other looks at species interactions using trait information. The latter shows that the fruit size range of palms dispersed by mammals is much greater than that of birds.

Line 292-294: I find this sentence confusing. Are you saying that it was more likely for plants to go extinct than adapt (evolve smaller fruits)?

Thank you for this comment. We realize that our sentence was a bit misleading with our original wording. In the macroevolutionary study cited (i.e., Onstein et al 2018), they show using a time-dependent speciation-extinction phylogenetic model, that among New World palms, large-fruited lineages have experienced higher extinction rates than small-fruited lineages through the Quaternary. The sentence has now been re-written for clarity.

Line 314: You can remove the “thus” since you already used “since” earlier in the sentence.
Corrected.

Conclusions

Lines 343-346. This is a really long somehow confusing sentence. It also may need citations
Agreed. Some citations have been added, and previously long sentences have been split up.

Figures

Figures are really well made, neat and clear. It is also appreciated that they chose a color-blind friendly scheme.

Figure 1: Define present-natural in figure caption as well

Very good idea. A definition for both “current” and “present-natural” scenarios have now been included in the caption.

Comments from response to reviewers

This is my take on the answers to Reviewer2’s concerns.

Model averaging issue

I’m not up-to-date with model-averaging literature, but I don’t think they do a good job at justifying its use. With so much literature on the topic, a better explanation should be provided beyond Reviewer2’s concerns. Additionally, when they mention “fitting a set of candidate models”, I can’t find what models are in this “set”. So overall, I think this section needs more work. I’m sure they can find literature to justify the use of model averaging and keep analysis as they are.

We now include additional justification for our model averaging approach and clarified our explanation of what is in this model set in the Methods to prevent the Results from being too long.

“We used a multi-model averaging approach to estimate effect sizes for each predictor variable across a set of candidate models (Burnham & Anderson). The primary advantage of model-averaging is that it accounts for model uncertainty across a set of candidate models, unlike more traditional stepwise model selection approaches that attempt to identify the single best model (Grueber et al 2011, Harrison et al 2018). We defined our candidate models as the set of ordinary least squares (OLS) linear regression models with log-transformed palm fruit size as the response variable, with each model containing a different combination of log-transformed frugivore body size, present-day climate and past climate change as covariates.”

Use of ordination analyses

I agree with Reviewer2 that ordinating a subset of variables only might not be the best approach. Alternatively, I propose two fast-achieving analysis. One in which you ordinate ALL variables together as proposed by Reviewer 2 to test whether results are comparable. Another in which you do NOT ordinate any of the variables and apply model-averaging to all variables, including each of the 6 climate variables. This does not need to be in the main manuscript. Just mention it in the text and include it in supplementary materials. It is likely that more readers come up with the same issue.

Both suggested analyses are problematic. The reason why we only ordinate the climate variables is that we are not primarily interested in which specific aspects of climate may drive variation in fruit size, but how much variation in climate as a whole may play a role. We have added further justification for our decision in the Methods (pasted below):

“However, because some of these bioclimatic variables are highly collinear with each other and as we were primarily interested in the relative importance of current climate and not the influence of specific climatic variables per se, we summarized the variation across the six bioclimatic variables using a principal component analysis (PCA).”

This approach allows us to separate the effect of present-day climate when interpreting the relative role of body size and past climate change. This is not possible if all variables are combined into one ordination

(the axes will reflect variation across multiple variables and axes) and the results of such an analysis cannot be directly compared with our approach (i.e. the coefficients of body size vs. a PCA axis incl. body size would not be comparable). Model coefficients of ordinated axes would also be difficult to interpret, as they explain variation across multiple different variables. We are therefore not convinced that this approach is meaningful.

Regarding the second suggestion of not ordinating any variable: This is also highly problematic due to multicollinearity among variables. In fact, we have tried to implement the suggested analysis but the models failed to converge, as some of the original bioclimatic variables are highly collinear. Thus, we are very concerned about following this suggestion because it violates the most fundamental assumptions of multivariate regressions.

That said, if the concern of the reviewer is the reliability of effect sizes because of collinearity among the ordinated and unordinated variables in our models, we can confirm that this is not the case as our main results do not change qualitatively if we standardize the coefficients by their partial standard deviations prior to model averaging. This has been recommended by Cade (2015 *Ecology*) and reviewed in Harrison et al. (2018 *Peer J*) as a statistical technique to control for collinearity on model averaged coefficients. We therefore show that our results are not affected by collinearity by ordinated (climate) and unordinated (body size, past climate) variables, while at the same time keeping the interpretability of the relative effect size (stand. coeff.) of body size on palm fruit size. We now also refer to correlograms showing the degree of correlation between predictor variables in the supplementary (Supplementary Figure 3) in the Methods as well. We have also further justified our approach in the Methods (Line XX).

Gbif data

I understand the author's concerns about GBIF data, but I also agree it would be informative to see the same analysis being performed with GBIF data. I don't think they need to do the analysis, but include more justification in the text/supplements. They do a really good job in the response, use it.

We now further illustrate the taxonomic incompleteness of GBIF data with some statistics we generated for our initial response: that 30% of palm species are not represented at all among records accessed through GBIF. This is in the 1st paragraph of the Methods.

Frugivores

Reviewer 2 raises an important point, something I would have been concerned about as well from the earlier version of the manuscript. I like that they repeated the analysis with what they call a "more conservative" definition. I'm not sure I would describe the former as a "conservative" though.

We understand the reviewer's point that our conservative definition is not really "conservative" and perhaps even better justified than the other. We therefore changed the terminology (but kept the definitions as such) and now refer to the results under the original "conservative definition" as the default. Our original default results are now referred to as those under a "liberal" classification, whereas our "more conservative" definition is now simply referred to as the "conservative" definition. The new definitions are now reflected in our Methods.

All figures in the main manuscript and supplement have now been updated to reflect these terminological changes, but the results as such remain the same.

When it comes to fossil species, a simple literature review could shed some light on whether considering some of the mammals frugivores or not. For instance, we know about giant sloth diet from their coprolites.

If you treated differently animals for which modern analogs would be poor proxies, you could potentially do a literature search for them. This would make the justification for including them stronger.

The reviewer suggests that we perform a literature review, which we in fact had already performed and underlies the classification of extinct mammals under our "Conservative" frugivore definition (see original supplementary information). To better clarify this, we now better describe how we used the paleontological literature in our frugivore definition in the Methods (see below)

We classified Pleistocene taxa as frugivores using three criteria: 1) if the taxon was predominantly herbivorous, 2) if the taxon belonged to families where $\geq 50\%$ of extant species are predominantly frugivorous, and 3) where the taxon either does not come from an extant families or orders (e.g., litopternids and toxodons) or where extant taxa may be poor analogs for diet for extinct taxa (e.g., sloths), if available paleontological evidence (e.g., isotopic, dental microwear etc) suggests the taxon may have been a browser or mixed-feeder. To evaluate the sensitivity of our frugivore classification, we additionally consider a more "liberal" classification of frugivory where we consider all predominantly herbivorous taxa as potential frugivores (i.e., only criterion 1), and a more "conservative" classification where only predominantly herbivorous taxa from pre-dominantly frugivorous extant families are included (i.e., criteria 1 and 2). A more detailed explanation of each definition is provided in the Supplementary Information.

Extinction risk

I think they did a good job addressing concerns.

Thank you, we're glad to hear that, and thank Reviewer 2 for the suggested improvements and Reviewer 3 for evaluating our improvements.

Megafaunal cutoff

I am not concerned about them using the 1,000Kg cutoff, but I'd appreciate a justification in the text if they are not using the traditional 44Kg. Choosing the number was not as arbitrary as they make it sound.

There is a large amount on paleontological research on the matter.

We do not use any cut-off in our analyses, but we understand that people might have a very different reference point when they come across the term "megafauna". We now define the term in the introduction with references to the traditional 44 kg (100 lbs) and include a citation.

Reviewers' Comments:

Reviewer #3:

Remarks to the Author:

I appreciate the authors took their time addressing my concerns. The newer version looks even stronger and I don't have any other comments.

REVIEWERS' COMMENTS:

Reviewer #3 (Remarks to the Author):

I appreciate the authors took their time addressing my concerns. The newer version looks even stronger and I don't have any other comments.